# Affinity-matured homotypic interactions induce spectrum of PfCSP structures that influence protection from malaria infection

Gregory M. Martin[1], Jonathan L. Torres [1], Tossapol Pholcharee [1,7], David Oyen[1,8], Yevel Flores-Garcia[2], Grace Gibson[1], Re'em Moskovitz[1], Nathan Beutler [3], Diana D. Jung[1], Jeffrey Copps [1], Wen-Hsin Lee [1], Gonzalo Gonzalez-Paez[1], Daniel Emerling [4], Randall S. MacGill [5], Emily Locke[5], C. Richter King[5], Fidel Zavala [2], Ian A. Wilson [1,6] & Andrew B. Ward [1] ✉

The generation of high-quality antibody responses to *Plasmodium falciparum* (Pf) circumsporozoite protein (PfCSP), the primary surface antigen of *Pf* sporozoites, is paramount to the development of an effective malaria vaccine. Here we present an in-depth structural and functional analysis of a panel of potent antibodies encoded by the immunoglobulin heavy chain variable (IGHV) gene *IGHV3-33*, which is among the most prevalent and potent antibody families induced in the anti-PfCSP immune response and targets the Asn-Ala-Asn-Pro (NANP) repeat region. Cryo-electron microscopy (cryo-EM) reveals a remarkable spectrum of helical antibody-PfCSP structures stabilized by homotypic interactions between tightly packed fragments antigen binding (Fabs), many of which correlate with somatic hypermutation. We demonstrate a key role of these mutated homotypic contacts for high avidity binding to PfCSP and in protection from Pf malaria infection. Together, these data emphasize the importance of anti-homotypic affinity maturation in the frequent selection of *IGHV3−33* antibodies and highlight key features underlying the potent protection of this antibody family.

Vaccines are critical tools for the sustainable elimination of malaria, which in 2020 was responsible for 241 million infections and 627,000 deaths worldwide (World Malaria Report[1]). The pressing need for an improved vaccine is underscored by the continual emergence of resistance to antimalarial compounds by the malaria parasite, *Plasmodium falciparum* (Pf)[2]. In an important milestone for global health, the first vaccine for malaria, RTS,S/AS01 (RTS,S), received a recommendation for widespread use in young children living in areas of moderate to high *P. falciparum* malaria transmission by the World

Health Organization (WHO) in late 2021. However, the initially robust immune response and protective efficacy conferred by RTS,S are transient, as both wane rapidly after about one year. Thus, a key challenge in malaria vaccine design is the generation of highly effective and long-lived (durable) immunity.

Many malaria vaccine candidates, like RTS,S, are based on *P. falciparum* circumsporozoite protein (PfCSP), which is the primary surface antigen of *P. falciparum* sporozoites, the stage of malaria parasites infectious to humans. The structure of PfCSP comprises three domains

[1]Department of Integrative Structural and Computational Biology, The Scripps Research Institute, La Jolla, CA 92037, USA. [2]Department of Molecular Microbiology and Immunology, Malaria Research Institute, Johns Hopkins Bloomberg School of Public Health, Baltimore, MD 21205, USA. [3]Department of Immunology and Microbiology, The Scripps Research Institute, La Jolla, CA 92037, USA. [4]Atreca Inc, San Carlos, CA 94070, USA. [5]PATH's Malaria Vaccine Initiative, Washington, DC 20001, USA. [6]The Skaggs Institute for Chemical Biology, The Scripps Research Institute, La Jolla, CA 92037, USA. [7]Present address: Department of Biochemistry, University of Oxford, Oxford OX1 3DR, UK. [8]Present address: Pfizer Inc, San Diego, CA 92121, USA. ✉e-mail: andrew@scripps.edu

(Fig. 1): (1) a flexible N-terminus, which contains a heparin sulfate binding site for hepatocyte attachment; (2) a central repeat region composed of 25 to 40 major (NANP) repeats, which are interspersed by a few, N-terminal minor repeats (NVDP, NPDP); and (3) a small, structured C-terminal domain. Vaccination with whole sporozoites or full-length PfCSP generates antibodies against each domain, but the NANP repeats are immunodominant[3-5]. Moreover, anti-NANP monoclonal antibodies (mAbs) have been shown to confer sterile protection against malaria infection in animal models through their ability to arrest sporozoite motility in the skin and to block liver infection[6-11].

Early observation of these effects provided the rationale for the design of RTS,S (Fig. 1), a virus-like particle based on the Hepatitis B surface antigen (HBsAg) that displays 19 NANP repeats and the ordered C-terminal domain of CSP[12]. Phase III clinical trials have shown that, in children aged 5–17 months, RTS,S confers modest protection (~50%) from clinical malaria at 12 months after the third vaccine dose[13], which waned to 26% at 4 years in follow-up studies[14]. Anti-NANP titers are associated with protection[15], and display similar induced antibody decay kinetics to other vaccines following vaccination[16]. Thus, improving vaccine efficacy requires boosting antibody quantity over time (durability) and/or improving antibody quality (potency).

A modern approach to vaccine design entails structural analysis of potent monoclonal antibodies (mAbs) in complex with antigen[17]. To this end, recent X-ray and cryo-EM structures have shown the repeat region is organized into NPNA structural units[18-22], and that the NPNA prolines serve as key anchor points for conserved aromatic residues in the heavy and light chain CDR loops. Interestingly, the humoral response to PfCSP is heavily biased towards antibodies descended from the human heavy chain germline gene *IGHV3–33*[19,20], which has also given rise to the majority of the most potent anti-NPNA mAbs isolated to date.

We previously showed one such highly potent *IGHV3–33* mAb, mAb 311, utilizes homotypic interactions to stabilize an extended helical structure of 311 Fabs bound to rsCSP, which is a recombinant form of PfCSP containing 19 NANP repeats (Fig. 1)[23]. Somatically mutated residues mediating key homotypic contacts between adjacent Fabs were critical for the stability of the extended helical structure but were not directly involved in CSP binding. Homotypic contacts were also observed in the structures of two other potent *IGHV3–33* mAbs 1210 and 239[19,22]. Interestingly, in mAb 1210, mutations designed to disrupt these contacts significantly reduced B-cell activation in response to NANP$_5$, without substantially impacting affinity to NANP$_3$, implying that homotypic interactions may occur in vivo between adjacent B-cell receptors in response to CSP antigens. Overall, these observations suggest the nature of the NANP repeats facilitates antibody-antibody (anti-homotypic) affinity maturation, which may underlie the frequent selection of the *IGHV3–33* germline. However, whether homotypic interactions contribute to the protective efficacy of soluble antibodies, and if they occur on the surface of sporozoites, has not been demonstrated. To address these questions, we expanded our investigation of *IGHV3–33* mAbs[22], and used electron microscopy combined with in vivo and in vitro assays to understand the structural basis of CSP engagement by this family of mAbs, the role of homotypic interactions, and the mechanism of protection from malaria infection.

## Results
### Helical structure formation on CSP is common among *IGHV3–33* mAbs
The antibody sequences in the current study were isolated from protected individuals within the dose fractionation arm of a Phase IIa clinical trial of RTS,S[24], the same trial from which mAbs 311 and 317 were derived, and were positively selected from a larger subset based on binding to PfCSP and NANP$_6$ by ELISA[24]. We focused specifically on antibodies encoded by the heavy chain germline gene *IGHV3-33*, which has given rise to many potent anti-NPNA mAbs with a

tendency toward homotypic interactions, as exemplified by Abs 1210 and 311[19,23]. In all, the panel includes seven *IGHV3-33* mAbs (Table 1), including 311 for comparison, which are encoded by three different light chain genes: *IGKV1-5* (mAbs 239, 334, and 364), *IGKV3-15* (mAbs 337 and 356), and *IGLV1-40* (mAbs 311 and 227).

To structurally characterize the interaction of each mAb with PfCSP, we formed complexes of the Fabs with rsCSP (Fig. 1). Initial negative-stain electron microscopy (NS-EM) imaging showed each *IGHV3–33* Fab formed well-ordered, multivalent structures on rsCSP (Supplementary Fig. 1), with well-resolved Fabs radiating outwards from a central rsCSP polypeptide. For comparison, we performed the same analysis with a panel of non-*IGVH3–33*-encoded mAbs (*IGHV3–30, 3–49, 3–15,* and *1-2*) isolated from the same clinical trial (Supplementary Fig. 1); we previously published EM data on two of these: 317 and 397[18,21]. Similarly to the *IGHV3–33* mAbs, the non-*IGHV3–33* panel bound multivalently to rsCSP and displayed general helical or spiral curvature. However, the 2D NS-EM classes demonstrate greater structural variation and the absence of long-range helical order. Accordingly, we were unable to obtain stable 3D reconstructions from NS or cryo-EM of these non-*IGHV3–33* Fab-rsCSP complexes. These EM data suggest that among human mAbs, long-range helical structural ordering stabilized by homotypic interactions may be specific to the *IGHV3–33* germline.

### *IGHV3–33* mAbs exhibit a spectrum of helical forms on rsCSP
We next utilized single particle cryo-EM to elucidate the 3D organization of these distinctive Fab-rsCSP structures, the potential roles of homotypic contacts, and the mechanisms governing their formation. Cryo-EM datasets were collected for the seven *IGHV3–33* mAbs in our panel. Each complex was resolved to high resolution (Supplementary Table 1), and the cryo-EM maps are shown in Fig. 1. Of these, six structures are new, while the 311-rsCSP structure was re-refined from our previously published cryo-EM dataset to achieve higher resolution (3.0 Å from 3.4 Å). In general, each complex was homogeneous in both structure and composition, with the overall resolution of the reconstructions ranging from 2.7 to 3.8 Å.

This compendium of high-resolution structures reveals the remarkable conformational plasticity of PfCSP. In each complex, CSP displays some form of helical structure that is stabilized by homotypic interfaces between tightly packed Fabs bound along the length of the NPNA repeats. However, the observed helical conformations of CSP vary dramatically. These range from near-planar discs with a shallow pitch and large helical radius, as observed in 364 and 227, to extended helices with varying helical parameters (Table 1). Each of the extended helices in complexes with 337, 334, 311, 356, and 239 are right-handed, while the partial, disc-like helices of 364 and 227 display left-handed curvature. The extended helical structures of CSP in the 337, 334, and 239 complexes are each unique, i.e., non-superimposable. Strikingly however, the 311 and 356 rsCSP helical structures are almost perfectly superimposable (Supplementary Fig. 2), which is notable as these mAbs utilize distinct homotypic interactions and different light chains (*IGLV1–40* and *IGKV1–5*, respectively; Supplementary Fig. 3). This finding suggests that either this is a relatively stable conformation of PfCSP, or that this particular structure is associated with high-level protection from malaria infection, as both 311 and 356 have been shown to be highly protective in in vivo mouse challenge models[22].

The 227 Fab complex is distinct, as the NPNA repeats form two discontinuous, anti-parallel disc-like structures with moderate helical pitch and left-handed curvature, with each disc bound by 4 Fabs in tandem (Fig. 1). However, we note the 227 Fab structure was solved in complex with NPNA$_8$ peptide instead of rsCSP, as was done for the rest of the mAbs in the panel, due to the tendency of the 227-rsCSP

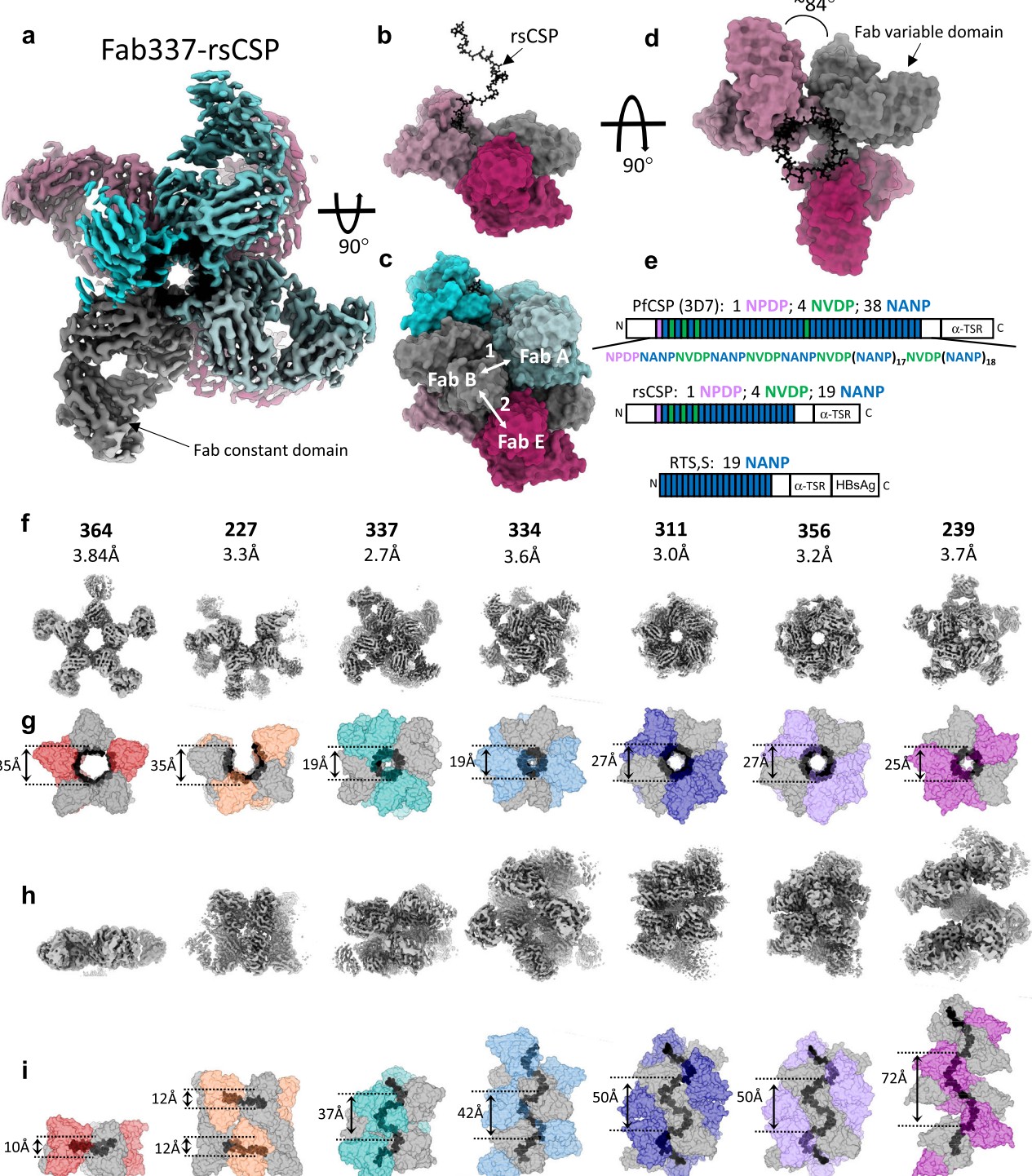

**Fig. 1 | High resolution cryo-EM of *IGHV3–33* Fabs in complex with rsCSP. a** Cryo-EM map of 337-rsCSP at 2.7. Å, viewing down the axis of the rsCSP helix. **b** Side-view of the 337-rsCSP structure, with four of seven Fabs removed to highlight rsCSP helical structure in black. In all structures, the Fab constant domain is disordered, thus only the Fab variable region is modeled. **c** Same as in (b), with all seven Fabs shown. Two homotypic interfaces (1 and 2) are highlighted. **d** Top view of b. Rotation angle between Fabs (helical turn) is shown. **e** Schematic of PfCSP sequences relevant to current study. **f** Top view, i.e., as viewed down the axis of rsCSP helix, of cryo-EM maps. mAb name and the resolution of each cryo-EM map are listed. In panels f-i, all structures and maps are on the same scale to enable comparison of relative dimensions. **g** Top view of the molecular surface representation of the various structures. rsCSP is colored in black. Diameter of the rsCSP helix is listed. **h** Side view of the cryo-EM maps. **i** Side view of the cryo-EM structures, displayed as a molecular surface. Helical pitch is shown.

complex to aggregate. Thus, the two antiparallel disc structures in the 227 complex likely comprise two individual NPNA$_8$ peptides, as the four available NPNA$_2$ epitopes on the peptide are fully occupied by 227 Fabs and there is no density linking the two discs. Nonetheless, a NS-EM reconstruction of the 227-rsCSP complex is nearly identical to the 227-NPNA$_8$ cryo-EM structure, within the 15–20 Å limit of the NS data (Supplementary Fig. 4). Therefore, this antibody may induce dimerization of separate CSP molecules mediated by homodimeric interactions of the Fabs themselves, which may have important implications for the way this antibody engages PfCSP on sporozoites.

**Table 1 | Structural features of antibodies in this study, and helical parameters of Fab-rsCSP cryo-EM structures**

| | | | CDR3 Length (aa) | | SHM (# aa) | | | Helical Parameters | | | | | | |
|---|---|---|---|---|---|---|---|---|---|---|---|---|---|---|
| mAb | HV Allele | KV/LV Allele | HC | LC | HC | LC | Full epitope (cryo-EM) | Map res. (Å) | Fabs bound | Turn (°) | Fabs/turn | Rise (Å) | Pitch (Å) | Diam (Å) |
| 227 | 3-33*01 | LV1-40*01 | 15 | 11 | 10 | 3 | A**NPNANPNA** | 3.3 | 8 | 69.9 | 5.2 | 2.3 | 12 | 35 |
| 311 | 3-33*01 | LV1-40*01 | 12 | 12 | 10 | 5 | A**NPNANPNA** | 3.0 | 11 | 68 | 5.3 | 10.6 | 50 | 27 |
| 239 | 3-33*03 | KV1-5*05 | 12 | 10 | 11 | 9 | Npn**ANPNANPNA** | 3.72 | 10 | 71.9 | 5 | 14.4 | 72 | 25 |
| 334 | 3-33*08 | KV1-5*05 | 14 | 10 | 10 | 6 | Npn**ANPNANPNA** | 3.62 | 9 | 80 | 4.5 | 9.3 | 42 | 19 |
| 364 | 3-33*03 | KV1-5*05 | 10 | 8 | 10 | 8 | A**NPNANPNA** | 3.84 | 5 | 61.7 | 5.8 | 1.7 | 10 | 35 |
| 337 | 3-33*08 | KV3-15*01 | 15 | 8 | 12 | 4 | A**NPNANPNA**NP | 2.7 | 7 | 83.8 | 4.3 | 8.6 | 37 | 19 |
| 356 | 3-33*03 | KV3-15*01 | 15 | 8 | 12 | 8 | A**NPNANPNA**NP | 3.2 | 11 | 68 | 5.3 | 10.6 | 50 | 27 |

Germline alleles were derived from the IMGT database, and CDR lengths are according to IMGT definitions. Full epitope describes the complete epitope of one Fab in the rsCSP cryo-EM structure. The core NPNA$_2$ epitope is in bold. Map res. is the overall resolution of reconstruction. The helical parameters were calculated from measurements in UCSF-Chimera. *Helical turn*: the angular step between adjacent Fabs on central rsCSP helix, as measured from the center of the helix. *Helical pitch*: the length required to complete one full helical turn, measured parallel to the rsCSP helical axis. *Helical rise*: the distance traversed along the rsCSP helix by each Fab, measured parallel to helical axis.

## The *IGHV3–33* NPNA$_2$ core epitope structure is highly conserved

As shown previously for this family of antibodies, the epitope of each *IGHV3–33* mAb comprises two tandem NPNA structural units, with an N-terminal type 1 β-turn followed by an Asn-mediated pseudo $3_{10}$ turn[18,20,22,23]. Interestingly, despite large differences in global helical structure, the local structure of this core (NPNA)$_2$ epitope is highly conserved and exhibits a nearly identical extended S-shaped conformation in each of the seven mAbs (Fig. 2b). rsCSP binds within a deep groove running along the length of each Fab that is composed entirely of the three heavy chain CDR loops and CDRL3 (Fig. 2a). Overall, the structure of the *IGHV3–33* heavy chain is also highly conserved. Moreover, the cryo-EM structures of Fabs of 239, 356, and 364 correspond very well to our previously determined X-ray structures of these three Fabs in complex with NPNA$_2$ (Supplementary Fig. 2)[22].

As noted previously for mAbs 311, 239, 356, and 364, conserved aromatic residues in CDRH2 of mAbs 227, 337, and 334 also each utilize the two prolines of the NPNA$_2$ epitope as anchor points (Fig. 2b–d). The strictly conserved, germline-encoded W52 and either a Tyr (germline) or Phe at position 58 (Y/F58) each form critical, alternating CH-π interactions with the Pro of the pseudo-Asn $3_{10}$ turn and the type 1 β-turn, respectively (Fig. 2d). These two NPNA structural units reside in two distinct hydrophobic pockets in each Fab (site 1 and site 2). While each Fab binds CSP through differing sets of interactions, this basic paratope architecture is conserved across the panel: site 1, which binds the type 1 β-turn, comprises residues from CDRH2 (Y/F58), CDRH3, and CDRL3; and site 2, which binds the pseudo-Asn $3_{10}$ turn, is formed from the three HCDR loops and is centered on W52 and another conserved aromatic residue in CDRH2, Tyr/His52A, which in each structure packs tightly against the side chain of the C-terminal Ala of NPNA$_2$ (Fig. 2c).

Importantly, the cryo-EM structures show the full epitope for a single Fab extends beyond NPNA$_2$, such that adjacent Fabs engage overlapping epitopes with between 1 and 4 shared residues at the N- and C-terminal ends of each NPNA$_2$ core (Fig. 2e–h; Table 1). The extent of the full epitope footprint on rsCSP tends to correlate with light chain usage and CDRH3 and CDRL3 length (Table 1). Thus, these key antibody features appear to determine the binding mode, superstructure assembly and fine epitope specificity of anti-NPNA antibodies and may also correlate with protective efficacy.

## A constellation of homotypic interactions stabilizes helical structures

Each of the multivalent antibody-CSP structures are stabilized by homotypic interactions between Fabs binding immediately adjacent NPNA$_2$ epitopes, i.e., the primary homotypic interface (Interface 1; Fig. 3). This expands the full paratope, as each Fab simultaneously binds both CSP and the neighboring Fab, and substantially increases the total buried surface area (BSA) on each Fab (Supplementary Table 2). The architecture of the primary homotypic interface is similar across the seven complexes and is composed mainly of the heavy chain CDR loops and CDRL3, with polar contacts between CDRH1$_B$-CDRH2$_A$ and CDRH3$_B$-CDRL3$_A$ (Fig. 3b). Importantly, this asymmetric, edge-to-edge interaction, in which FabA and FabB contribute different residues to the interface, is distinct from the asymmetric head-to-head configuration observed in the crystal structure of mAb 1210-NANP$_5$, another potent *IGHV3–33/IGKV1–5* antibody[19]. This mode of binding also differs from our previous crystal structure of 399-NPNA$_6$ (*IGHV3-49/IGKV2–29*), which forms a *symmetric* head-to-head homotypic interface between adjacent Fabs[22] (Supplementary Fig. 5). As these latter two mAbs are not known to form stable structures on extended repeats, the edge-to-edge binding mode seen here is likely necessary for optimal geometry and packing of Fabs to promote long-range helical order.

Homotypic interactions within the primary interface are derived from a diverse set of both germline-encoded residues and those that evolved through somatic hypermutation (SHM; Fig. 3f–i; Supplementary Fig. 6, Supplementary Tables 3–9). Two residues in CDRH1, T28 and S31, mediate key contacts between CDRH1$_B$-CDRH2$_A$ in nearly every complex in the panel (Fig. 3c–e, Supplementary Figs. 5, 6). T28 is a germline residue that is nearly strictly conserved (S28 in 337), while the S31N mutation is seen in four of the seven mAbs: 239, 311, 334, and 356 (Supplementary Fig. 3). Together, these residues coordinate an extensive network of hydrophobic and electrostatic interactions, with N31 often forming multiple critical contacts with evolved basic and aromatic residues in the neighboring CDRH2$_A$. Importantly, these specific interactions would likely not occur in the germline sequence (Fig. 4d–f). Other key residues in CDRH1$_B$ are R30 and F32, which in both mAbs 239 and 356 form a signature motif R$^{30}$N$^{31}$F$^{25}$, mutated from the germline sequence of S$^{30}$S$^{31}$Y$^{25}$ (Supplementary Fig. 3). In both structures, R30 forms a pair of hydrogen bonds with the N56 side chain and S55 main chain, both from CDRH2$_A$, while F32 forms an anion–π bond[25] with the evolved E64 in HFR3$_B$ (Fig. 3c). These interactions would also not occur upon germline reversion. Moreover, except for F32, none of the side chains of these residues directly contact CSP, providing evidence for affinity maturation to stabilize antibody-antibody rather than antibody-antigen interactions.

As previously shown for 311, the tight helical packing in the complexes of mAbs 334, 337, and 356 creates a secondary homotypic interface between Fabs separated by about one helical turn (3 or 4 NPNA$_2$ epitopes; Supplementary Fig. 7). Homotypic interactions

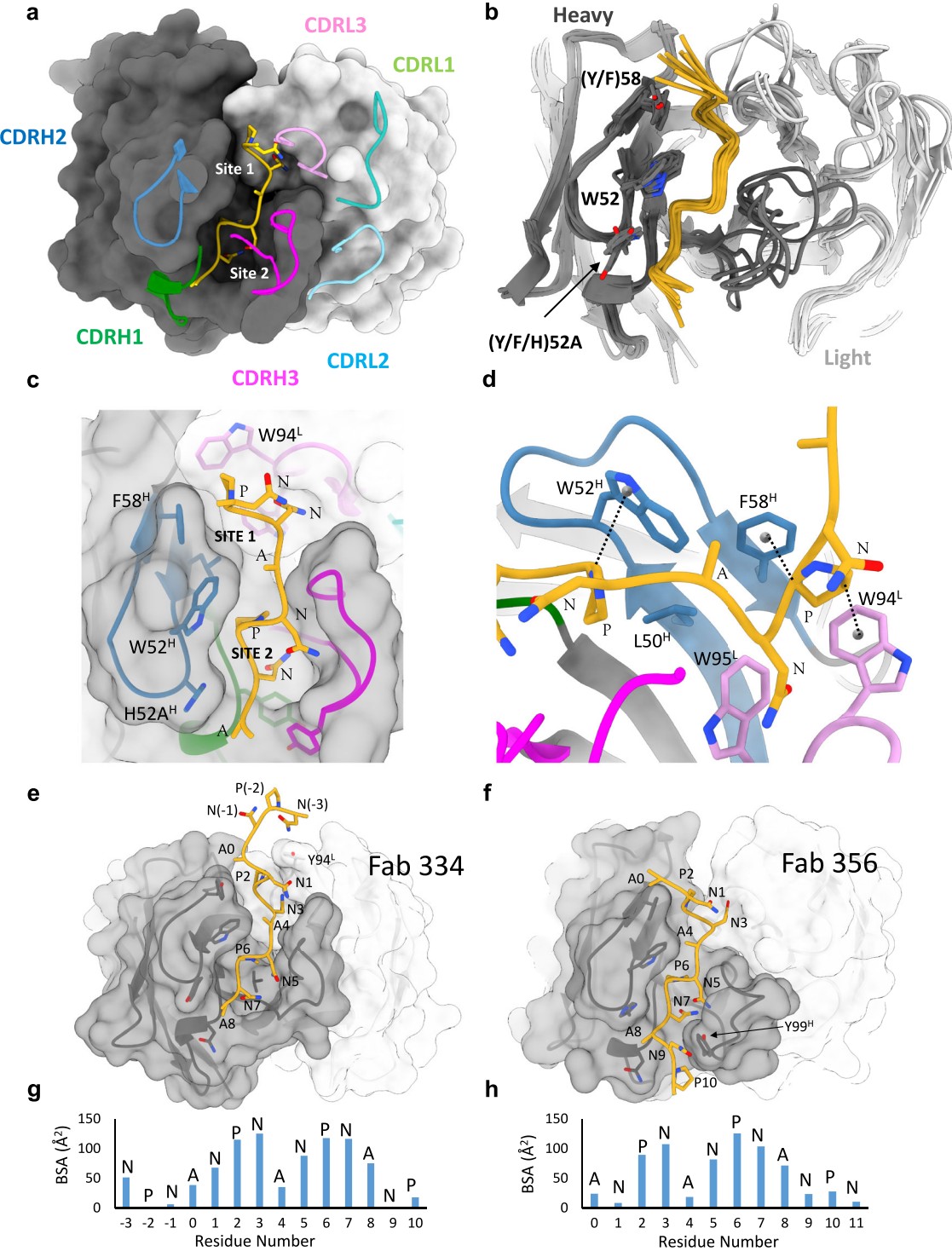

**Fig. 2 | Structure and conservation of the NPNA$_2$ epitope. a** Surface model of a single Fab from 337-rsCSP structure, showing only the core epitope NPNA$_2$ in gold. The heavy chain is colored dark gray, and light chain is in light gray. **b** Superposition of a single Fab and NPNA$_2$ from each of the seven cryo-EM structures. Same coloring as in a. **c** Zoomed-in view of the paratope of 337, highlighting two hydrophobic pockets, Site 1 and Site 2. **d** CH−π interactions of CDRH2 and CDRL3 residues with Pro in the NPNA repeat. **e**, **f** Full epitope structure of 334 and 356, showing N and C-terminal extensions beyond NPNA$_2$, which are labeled as residues 1−8. **g**, **h** Buried surface area (BSA) contributions by each residue within the full epitope of 334 (g) and 356 (h). See also Fig. S2. Source data are provided as a Source Data file.

consist mostly of polar contacts within apposing heavy and light chain framework regions (FRs; Fig. S6B–D). In contrast, the secondary homotypic interface in 227 mediates 227-NPNA$_8$ dimerization and defines the C2 symmetry plane for the complex (Supplementary Fig. 7B). This interface is therefore symmetric and consists solely of apposing heavy chain framework residues. In general, the secondary homotypic interface contributes about half of the total BSA relative to the primary interface (Supplementary Table 2). Strikingly, however, the reverse is true with 334, where the total BSA of the secondary interface is roughly twice that of the primary, suggesting a critical role for framework region residues in the stability and/or formation of this complex.

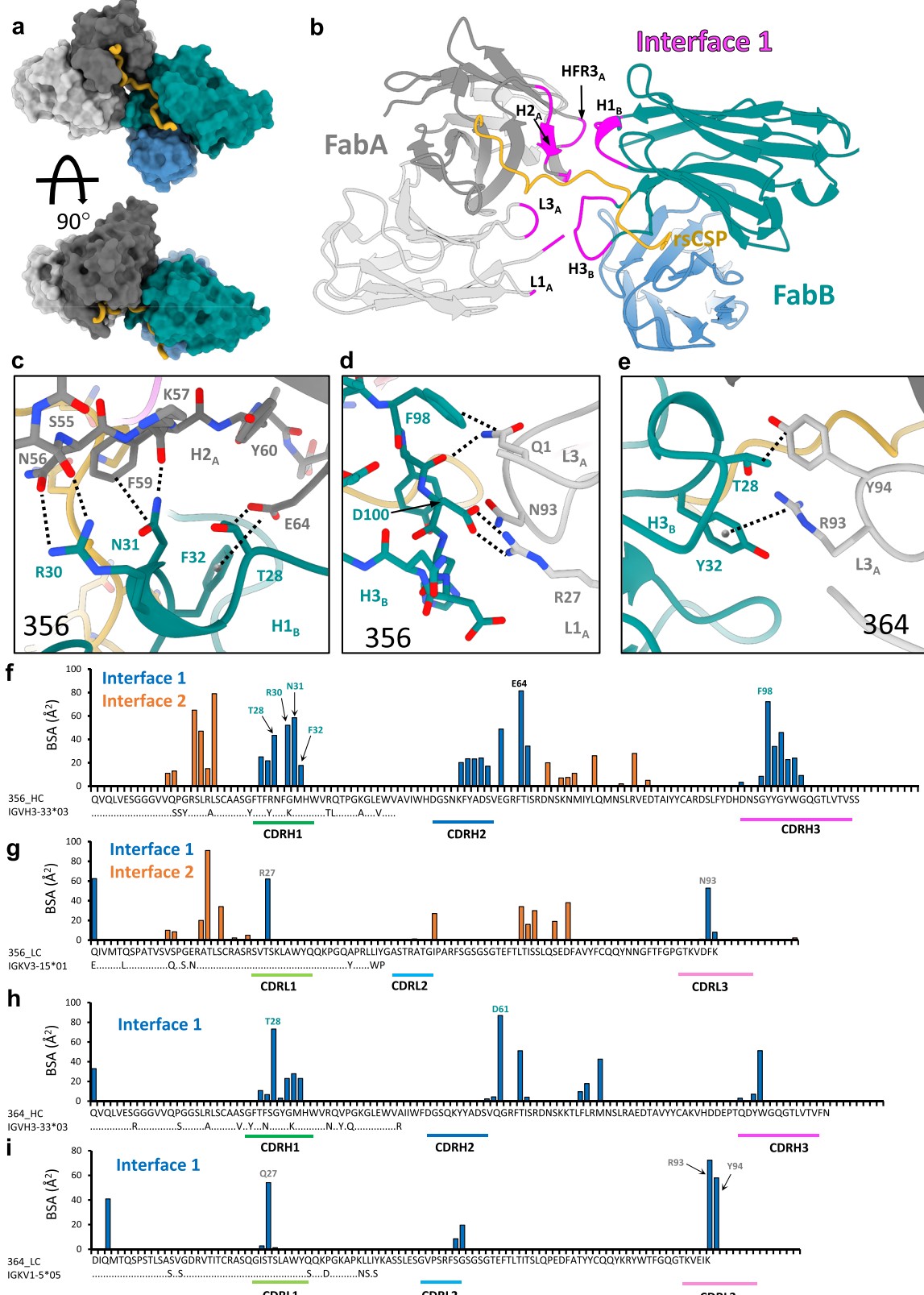

**Fig. 3 | Structure of the primary homotypic interface (Interface 1). a** Surface representation of two adjacent Fabs from 356-rsCSP structure. rsCSP is colored in gold. **b** Cartoon representation of same model as in (**a**), with residues mediating homotypic contacts highlighted in magenta. **c**–**e** Structural details of key homotypic interactions in 356 (**c**, **d**) and 364 (**e**). Specific contacts are indicated with dashed lines. **c** CDRH1 of FabB with CDRH2 of FabA. **d** CDRH3 of FabB with CDRL1 of FabA. **e** CDRH1 of FabB with CDRL3 of FabA. **f**–**i** Per-residue BSA contributions to homotypic interface identified in 356-rsCSP (**f**, **g**) and 364-rsCSP (**h**, **i**) structures. Note this plot does not contain BSA from CSP. See also Figs. S4, S5. Source data are provided as a Source Data file.

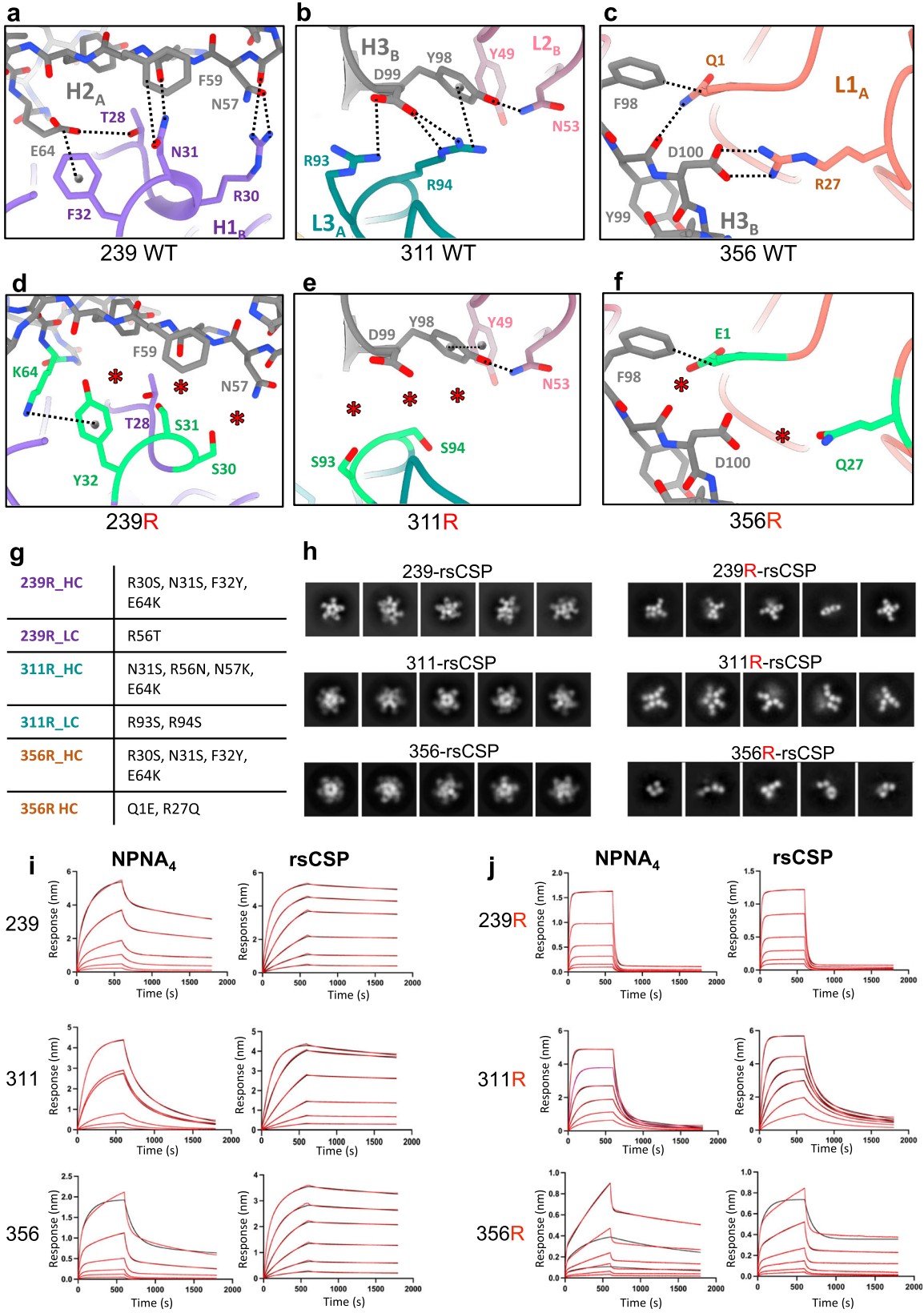

Key contacts within the secondary interface are also linked to somatic hypermutation of the germline heavy and light chain genes. In HFR1$_E$ of 334, a mutated residue T19 appears critical for the interface and mediates a key hydrogen bond with S65 of LFR3$_B$ (Supplementary Fig. 7D). In the symmetric secondary interface of 227, H82A of HFR3$_E$ mediates a cation-pi bond with R75 of HFR3$_B$ (Supplementary Fig. 7C).

Both were mutated from highly conserved residues in the germline *IGHV3−33* gene, (N82A-H and K75R). Moreover, the H82A-R75 interaction contributes nearly half of the BSA of this interface (250/550Å$^2$) and defines the C2 symmetry axis of the 227-rsCSP complex (Supplementary Fig. 7B, C), suggesting a critical role for this interaction. Overall, these examples represent somatic hypermutation in

**Fig. 4 | Structural and functional effects of mutagenesis of the homotypic interface. a–c** Key, somatically mutated homotypic interactions observed in cryo-EM structures of 239 (**a**), 311 (**b**), and 356 (**c**). Dashed lines indicate observed homotypic contacts. **d–f** Anticipated structural impact of reversion of these residues to germline identities. Mutant structures were calculated from WT cryo-EM structures in Coot and are not experimental. Red asterisk indicates loss of homotypic contacts. Dashed lines indicate potential germline-encoded homotypic contacts. **g** List of germline-reverted constructs. Mutations are listed on right, using Kabat numbering system. **h** 2D class averages from NS-EM of WT and mutant Fab complexes with rsCSP. Mutant classes on right clearly show loss of well-ordered helical structure observed with WT Fabs. **i, j** Binding curves from BLI for WT (i) and mutant (j) Fabs. NPNA$_4$ and rsCSP were immobilized on Streptavidin and Ni-NTA sensors, respectively, and binding of each of the Fabs were measured at 6.25, 12.5, 25, 50, 100, and 200 nM. Curves were fit with a 2:1 binding model shown in red. nm nanometers. See also Fig. S6. Source data are provided as a Source Data file.

framework regions distal from the antigen binding site and provide further evidence for antibody-antibody affinity maturation to enhance homotypic Fab–Fab interactions.

## Mutagenesis of the homotypic interface

We have previously shown germline reversion of the somatically mutated residues that mediate homotypic contacts, but which are not directly involved in CSP binding, abrogates the 311-rsCSP helical structure[23]. To further understand the role of homotypic contacts, we applied a similar approach to two other potent mAbs in our panel, 239 and 356. Our original mutant 311 construct, 311R, has four mutations in the heavy chain (N31S, R56N, N57K, E64K), and two in the light chain (R93S, R94S). To create both 239R and 356R constructs, the RNF motif in CDRH1 was mutated to germline, along with the same E64K mutation in HFR3 (R30S, N31S, F32Y, E64K). The light chains of 239R and 356R had one and two additional mutations, respectively (239R: R56T; 356R: Q1E, R27Q) (Fig. 4g).

We first determined the impact of these mutations on binding to various CSP peptides with biolayer interferometry (BLI). We tested the hypothesis that homotypic interactions underlie the large increase in apparent affinity to peptides with increasing NANP content that is observed for this family of antibodies. Thus, we compared the binding of WT and mutant Fabs to NPNA$_4$, NPNA$_8$, and rsCSP. The term "apparent affinity" is used to reflect the fact that peptides with four or more NPNA repeats contain multiple, non-independent Fab binding sites likely with multiple, non-independent affinity constants. In terms of NPNA$_4$, the apparent affinity of 311R was essentially unchanged relative to 311, while 356R and 239R were ~two-fold higher (improved) and ~two-fold lower than 356 and 239, respectively (Fig. 4i, j; Supplementary Table 10). These BLI data suggest that binding to minimal NPNA repeats is largely unperturbed by the germline mutations. As expected, for each WT Fab we observed a large increase in apparent affinity to both NPNA$_8$ and rsCSP relative to NPNA$_4$, largely driven by substantial reductions in the dissociation rate ($k_{off}$). However, for the reverted mutants, both affinity and $k_{off}$ remained roughly constant across each peptide and rsCSP. Thus, homotypic interactions are critical for high avidity binding to extended NANP repeats.

We next used NS-EM to assess the impact of the germline mutations on the structure of the Fab-rsCSP complex. As shown previously for 311R, 2D class averages of both 239R and 356R were highly variable, both in structure and stoichiometry of the Fabs (Fig. 4h). Interestingly, we observed some helical propensity in the 239R-rsCSP complex, similar to 311R, suggesting helical structure formation is at least partially germline-encoded or that CSP has a preferential bias toward a helical conformation. Nonetheless, we were unable to obtain stable 3D reconstructions for each mutant, indicative of a high degree of structural disorder. In contrast, the WT versions formed stable helical structures on rsCSP (Fig. 1, Supplementary Fig. 1). Thus, somatically mutated homotypic interactions are crucial for both high avidity and for the formation and stability of long-range, helical order on rsCSP, both of which may impact protective efficacy.

To ensure these effects were due to the loss of homotypic interactions rather than unanticipated changes in the structure of the antibody paratope, which could impact the structure of the bound NPNA$_2$ epitope, we solved a 1.9 Å co-crystal structure of Fab311R in complex with NPNA$_3$ and compared this to our previous X-ray

structure of Fab311 bound to NPNA$_3$ (Supplementary Fig. 8)[18]. Importantly, we find the structures of both Fab and CSP peptide are nearly identical, with an overall RSMD of 0.28 Å. Due to their similarity with 311R, we expect this to also be true for 239R and 356R, although we did not obtain crystals of these mAbs. Therefore, the effects of the germline mutations introduced here are likely confined to antibody-antibody binding with no significant impact on direct interactions with CSP, proving the usefulness of the germline-reverted mutants as tools to specifically probe the role of homotypic interactions.

## Affinity-matured homotypic contacts are important for protection

The role of homotypic contacts in protection from malaria infection is still unclear. To address this question, we compared the protective efficacy of WT and mutant 311, 239, and 356 using the liver burden assay (Fig. 5a), an in vivo model of malaria infection in mice that measures the ability of antibodies to prevent invasion of the liver by transgenic *P. berghei* sporozoites expressing *P. falciparum* CSP and luciferase[10,11].

Mice were injected intravenously (IV) with 75 μg of IgG (311, 311R, 239, 239R, 356, or 356R), and 16 h later were challenged with $2 \times 10^3$ transgenic sporozoites. Each mAb significantly reduced parasite liver burden relative to the naïve control (Mann–Whitney $U$ test; $p < 0.05$), which is reported as percent inhibition (Fig. 5b). Strikingly, however, 311R, 239R, and 356R each showed a consistent and dramatic reduction in percent inhibition relative to their WT counterparts, which was statistically significant in each case (Mann–Whitney $U$ test; $p < 0.001$). In a separate experiment conducted under near identical conditions, serum IgG concentrations were measured at the time of sporozoite challenge (16 h) and were similar across the WT and variant mAbs (Supplementary Fig. 9), which indicates that the differences in liver burden are likely due to differences in antibody interaction with sporozoites and not differences in antibody levels, in vivo mAb kinetics, or off-target responses. Overall, this is the first demonstration of a direct role of homotypic interactions in protection and implies these somatically mutated residues are critical for high-level protection from malaria, likely through their ability to mediate high avidity and helical structure formation with antibody–antibody homotypic interactions.

## Correlation of protection and affinity

We next compared the reduction in liver burden across each of the WT mAbs in our panel, using the same protocol as the previous protection experiment (Fig. 6a, b). For the mAbs with repeats across multiple experiments, i.e., 311, 239, and 356, and the highly-protective *IGHV3-30* mAb 317, the level of inhibition is consistent, enabling comparison of efficacy across separate experiments. As before, at 75 μg, each IgG significantly reduced parasite infectivity in the liver relative to the naïve control. While there is a range in the level of inhibition, many antibodies are highly potent and have statistically indistinguishable protection relative to mAb 317, namely mAbs 356, 311, 334, and 364. Protection for these mAbs generally ranges from around 85–92%. mAb 239 is slightly less potent than 317, at 81% inhibition, while 337 and 227 have the lowest levels of protection in the panel, at 68 and 52% inhibition, respectively ($p < 0.001$; Mann–Whitney $U$ test). However, the reduced potency of 227 may be due to poor

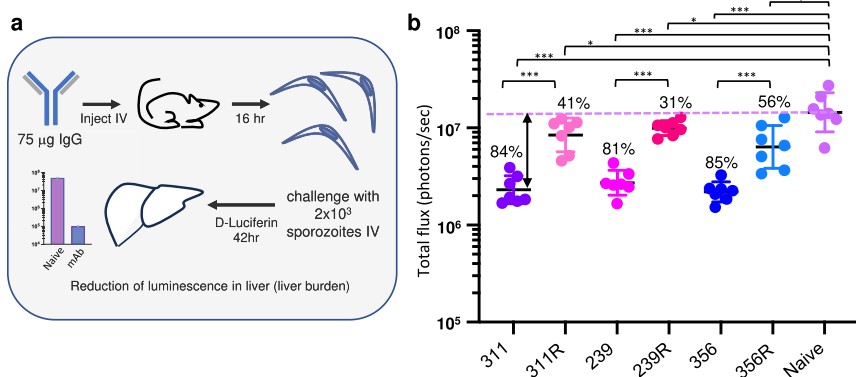

**Fig. 5 | Protective efficacy of WT and germline-reverted IgGs. a** Schematic of liver burden assay used to compare protective efficacy. **b** Liver luminescence measurements 42 h post-challenge; expressed as log total flux on the *Y*-axis. Each group (mAb) contained seven mice. Geometric mean and SD are indicated as black and colored lines, respectively. A two-sided Mann–Whitney *U* test was used to compare efficacy relative to naïve (no mAb) and between WT and mutant mAbs. Percent inhibition listed is relative to naïve. Note significant reduction in percent inhibition of each mutant mAb relative to the WT counterpart, indicating a key role of somatically mutated homotypic interactions in protective efficacy in this antibody family. Significance: *Naïve vs 239R, $p = 0.0379$; *Naïve vs 311R, $p = 0.0175$; *Naïve vs 356R, $p = 0.0111$; ***$p = 0.0006$. Adjustments were not made for multiple comparisons. For each group (antibody), $N = 7$ mice; data points represent individual mice. See also Fig. S7. Source data are provided as a Source Data file.

pharmacokinetics in vivo (Supplementary Fig. 9). Overall, these results are consistent with our previous liver burden data testing of many of these same mAbs at 100 μg[22].

As each of these mAbs target the same epitope(s) on CSP, affinity to the NPNA repeat may underlie differences in protection. To test this notion, we measured apparent affinities of each Fab to $NPNA_4$, $NPNA_8$, and rsCSP with BLI (Supplementary Fig. 10; Supplementary Table 10). Except for the germline mutants, apparent affinity increased substantially between $NPNA_4$ and $NPNA_8$, and again between $NPNA_8$ and rsCSP; this is likely due to the high avidity afforded by homotypic interactions, as increases in avidity were largely driven by reductions in the dissociation rate ($k_{off}$). In general, rsCSP apparent affinity was very high ($10^{-9}$M or higher) for the WT Fabs, and lower for the three mutants ($10^{-7}$ to $10^{-8}$M). We then correlated these data with percent inhibition from the liver burden experiment. Interestingly, we observe no correlation between protection and $NPNA_4$ affinity, while there is a moderate correlation with $NPNA_8$ and rsCSP apparent affinity ($R^2 = 0.61$ and 0.68, respectively), as well as rsCSP dissociation rate ($R^2 = 0.65$). These data suggest avidity to extended NPNA repeats, which is facilitated by homotypic interactions, is a key determinant of protective efficacy among *IGHV3–33* mAbs. However, apparent affinity to $NPNA_8$ or rsCSP poorly discriminates protective efficacy among the WT mAbs in the panel, which all have rsCSP apparent affinities of $10^{-9}$M or lower. Therefore, high avidity to extended repeats is likely necessary but on its own insufficient to confer high-level protection in *IGHV3–33* mAbs. Thus other parameters, likely concerning specifics of the interaction of antibodies with PfCSP on live sporozoites, also appear to be involved in determining protective efficacy.

## Discussion

The wealth of structural data presented herein, and the wide spectrum of observed helical conformations of rsCSP, are a vivid illustration of the extensive conformational plasticity of the NANP repeat region, which had been both predicted and demonstrated with indirect structural methods, but not directly or at high resolution[26–29]. Our panel of cryo-EM structures reveals how these diverse conformations are anchored by a subset of key, somatically mutated residues mediating homotypic interactions across two antibody-antibody interfaces, yet which do not directly participate in CSP binding. Intriguingly, we observe this behavior in each of the seven *IGHV3–33* mAbs we examined, suggesting that, within this antibody family, affinity maturation promotes the evolution of homotypic interactions that frequently lead to long-range, ordered helical structures on CSP.

Together, these data support a model in which the highly repetitive nature of the NANP repeats drives antibody-antibody affinity maturation, and that this selective advantage underlies the generation of high avidity antibodies and the frequent selection of the *IGHV3–33* germline.

We also demonstrate somatically mutated homotypic interactions, and possibly the CSP structures that they stabilize, play a key role in the mechanism of protection from *P. falciparum* infection. Specifically, we show reversion of these somatically mutated residues to their germline identities, in both heavy and light chains, abolishes well-ordered, extended helical CSP structures and eliminates the high avidity to extended NANP repeats characteristic of this antibody family, without significantly impacting affinity to the core epitope or the ability to assemble multiple Fabs onto CSP. Importantly, these effects are accompanied by a significant and consistent reduction in protective efficacy of the affinity-matured IgGs in vivo, relative to their WT counterparts, implying a critical role for homotypic interactions in protective efficacy for *IGHV3–33* mAbs.

Based on these data, we speculate that this family of IgGs bind multivalently on the surface of sporozoites in vivo, as Fabs do in vitro, and homotypic interactions that occur between adjacent IgGs and are critical for the stability of Ab-SPZ complexes and for protection. However, at present, little is known regarding the nature of the interaction between CSP antibodies and sporozoites, which makes it difficult to predict how differences in antibody structure or function may ultimately impact protective efficacy in vivo. Observations in this study and others indicate that NPNA affinity alone cannot fully account for protective efficacy[19,20,22]. Nevertheless, our functional and mutagenesis data strongly suggest high avidity to NANP repeats, driven by dramatic reductions in the off-rate, is a key component of antibody potency in the *IGHV3–33* family.

In terms of a vaccine design strategy, whether the development of homotypic interactions in an immune response is advantageous or not for both vaccine efficacy and durability remains to be determined. The current working hypothesis in the field posits that highly avid binding to extended NANP repeats, potentially afforded by homotypic interactions, induces strong B-cell activation but limits affinity maturation in germinal centers, which ultimately suppresses the development of antibodies with high affinity to the core NPNA epitope[30,31]. This would account for the robust antibody response to CSP, but also the difficulty in generating long-lived immunity and the generally low levels of somatic hypermutation observed in anti-NANP antibodies[32–34]. However, there is little direct evidence showing that CSP immunogens with

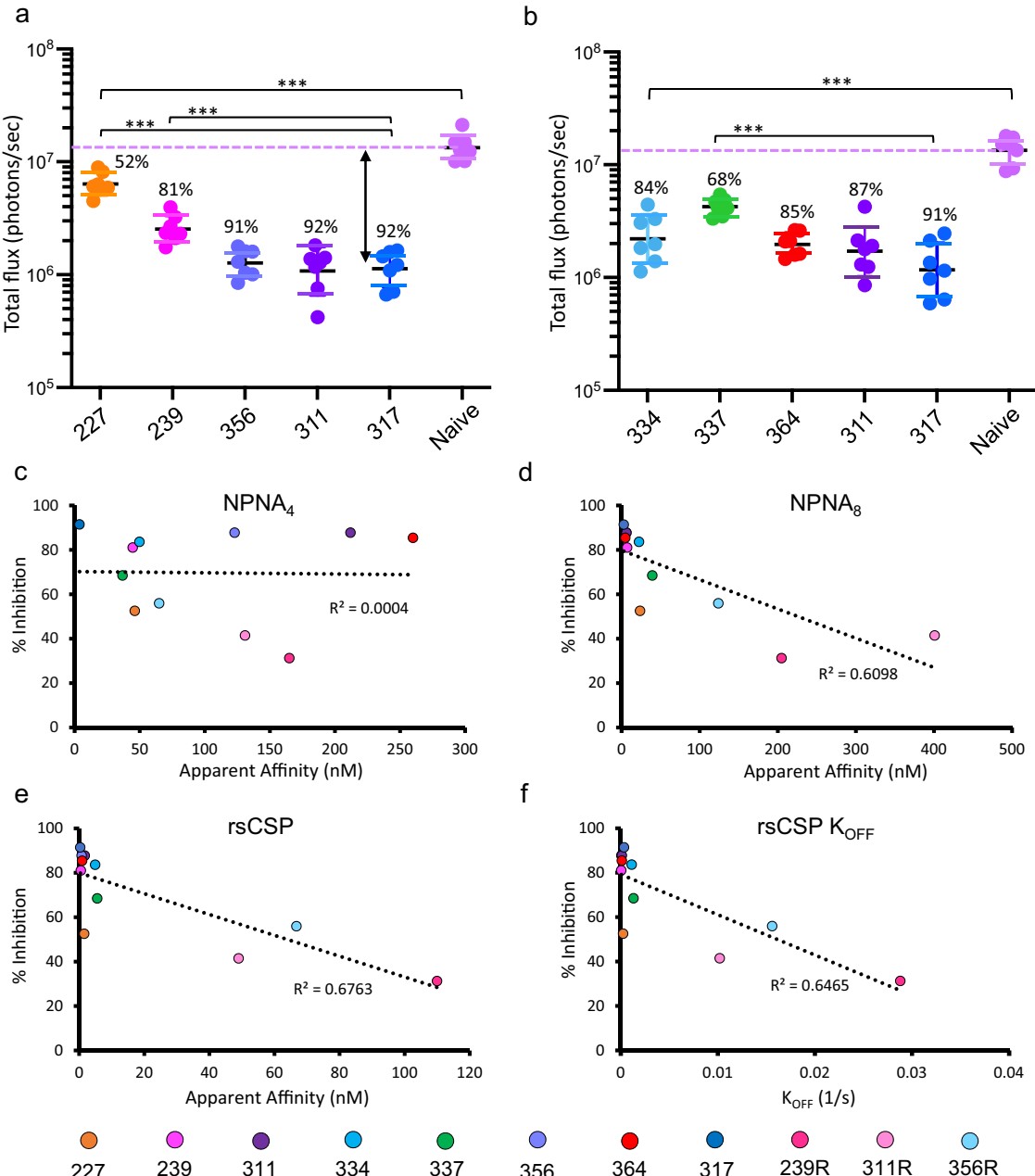

**Fig. 6 | Correlating protective efficacy and affinity to CSP. a, b** Liver burden results for each mAb in the panel. Two separate experiments were conducted, and mAbs 311 and 317 were included for comparison in each. Liver burden experiments were performed and analyzed as in Fig. 5b. Percent inhibition is relative to naïve (no mAb). A two-sided Mann−Whitney $U$ test was used to compare efficacy of reach mAb relative to naïve (no mAb). Significance: ***$p = 0.0006$. Adjustments were not made for multiple comparisons. For each group (antibody), $N = 7$ mice; data points represent individual mice. Error bars represent the geometric mean and SD. **c–e** Correlation of percent inhibition with apparent affinity of each Fab, as measured by BLI, to NPNA$_4$ (**c**), NPNA$_8$ (**d**), and rsCSP (**e**). Binding to immobilized NPNA$_4$ and rsCSP was measured at 6.25, 12.5, 25, 50, 100, and 200 nM, and binding to immobilized NPNA$_8$ at 12.5, 25, 50, and 100 nM. Binding curves were fit with a 2:1 model and affinity measurements were averaged across all fits ($\geq 4$) with $R^2 \geq 0.98$. **f** Correlation of percent inhibition with the rate of unbinding ($k_{OFF}$) from rsCSP in BLI experiments as in (**e**). See also Table S10. Overall, we observe no correlation of protection with affinity to NPNA$_4$, while we observe modest correlations to longer NPNA repeats (NPNA$_8$ and rsCSP). This indicates that, among *IGHV3–33* mAbs, binding avidity to extended repeats is a necessary but insufficient determinant of potency of protection. Source data are provided as a Source Data file.

reduced NANP content promote the development of higher affinity antibodies, and thus improved vaccine efficacy. While two studies have indicated a trend towards greater protection in mice immunized with constructs containing reduced numbers of NANP repeats (either 9 vs 27 NANP, or 5 vs 20 NANP), the results were not statistically significant[35,36]. Thus, future studies are needed to specifically assess the role of homotypic interactions in B-cell responses to CSP and whether

they underlie differences in immunogenicity to different repeat constructs.

Overall, our compendium of antibody-CSP structures provides a series of high-quality structural templates which will enable both structure-based vaccine design and antibody engineering. In particular, anti-PfCSP monoclonal antibodies have emerged as promising prophylactics for malaria, with two landmark studies demonstrating

the ability of two anti-PfCSP mAbs (cis43LS and L9LS) to provide months-long sterile immunity against controlled human malaria infection (CHMI) in humans[37,38]. While cis43 and L9 target different epitopes on PfCSP, namely the N-terminal junction and minor repeats, respectively, these human trials provide an important proof-of-concept that 1) mAbs identified as potent from mouse models of malaria infection are also highly potent in humans, and 2) that passive administration of potent anti-PfCSP mAbs is an effective tool for malaria prevention. Thus, a key aim is to identify the most potent mAbs and improve both their potency and pharmacokinetic properties through rational, in vitro and in vivo affinity maturation. The utility of this approach was demonstrated recently via CRISPR-based knock-in of cis43 germline heavy and light chain genes in mice[39], where the authors identified a cis43 derivative with greater potency than the best-in-class mAb L9.

Our panel of cryo-EM structures will also enable the rational design of NANP antigens that either promote or prevent the development of adjacent homotypic Fab–Fab interactions, or long-range bivalent IgG interactions, which may occur on extended NANP repeats. For example, based on our cryo-EM structures, a full helical turn encompasses a minimum of 10 and maximum of 12 NPNA repeats, as observed in 337-rsCSP and 239-rsCSP, respectively. Thus, a candidate antigen containing less than 10 NPNA repeats should prevent the development of the long-range homotypic interactions that stabilize extended CSP helices, which could impact the maturation pathway of induced antibodies. Conversely, the long-range CSP helical structure could be biochemically stabilized in a designed construct to specifically elicit highly potent 311, 356, or 239-like antibodies. Long-term, comparing a series of such antigens will be useful in parsing the potentially countervailing forces homotypic interactions may exert on vaccine efficacy. Moreover, in combination with the functional and in vivo protection data, these structural data enable the identification of the key structural, functional, and sequence-based features of highly potent anti-PfCSP antibodies, which at this point are still not well-defined.

The fact that each of the seven *IGHV3–33* mAbs we examined, many of which are highly potent, forms distinct, extended helical structures on rsCSP suggests that these EM structures may serve as structural correlates of protection. Critically, few clear correlates of CSP vaccine-induced immunity have been identified;[15,40] thus, EM-based analysis of antibody responses may be a powerful new tool for evaluating the efficacy of malaria vaccines. However, at present, the correlation of higher order structures with protective efficacy is not unequivocally resolved due to the small size of our antibody panel. Future studies with larger panels of monoclonals, and especially polyclonal serum from protected and nonprotected individuals, will determine whether this phenomenon is specific to human *IGHV3–33* mAbs, or whether it represents a general solution for a productive immune response to repeat antigens.

It is possible that the conformations of rsCSP captured here by cryo-EM correspond to distinct conformations of PfCSP on RTS,S particles or sporozoites, and that the maturation pathway of this antibody family proceeded through conformational selection of these different states. Intriguingly, while the number and positioning of the NANP repeats can vary across *P. falciparum* isolates, the amino acid sequence (NANP) is highly conserved[41], which suggests that the intrinsic disorder of the repeat region itself may serve as an immune evasion technique. Thus, the relative rarity of highly potent anti-PfCSP antibodies may be a function of the rarity of PfCSP conformational states competent to bind them. Further structural characterization of PfCSP, both in complex with antibodies and in the native context of sporozoites, will be critical for understanding the complex ways in which *P. falciparum* can interact with and evade the human immune system.

## Methods

### Ethics Statement

The studies involving mice were performed in strict accordance with the recommendations in the Guide for the Care and Use of Laboratory Animals of the National Institutes of Health. For liver burden experiments, the protocol was approved by the Animal Care and Use Committee of the Johns Hopkins University, protocol number MO18H419. For assessment of in vivo kinetics of IgGs, the protocol was approved the Institutional Care and Animal Use Committee (IACUC) at The Scripps Research Institute, protocol number 22-0007.

### CSP peptides

All peptides were produced by InnoPep Inc (San Diego, CA) at a purity level of ≥97%. Peptides for crystallography contained N-terminal acetylation and C-terminal amidation to eliminate charges at the peptide termini. Peptides for BLI were biotinylated at the C-terminus.

### Antibody sequences

All antibody sequences used in the current study were originally derived from the MAL071 clinical trial of RTS,S/AS01[24]. While none of this work was performed here, we note that the clinical trial protocol was approved by the Walter Reed Army Institute of Research (WRAIR) Institutional Review Board and the Western Institutional Review Board, and written informed consent was obtained from each subject before study procedures were initiated (Clinical Trials.gov identifier: NCT01857869). Plasmablast isolation and BCR sequencing of antibody genes in malaria vaccine trials have been previously described[24,42]. Fab or IgG1 heavy and light chain genes were codon-optimized and synthesized by GenScript (Piscataway, NJ).

### Protein production

Antibody genes were subcloned into pCMV or pCDNA3.4, either for expression as Fab or IgG1. Antibodies were expressed in ExpiCHO cells (Thermo Fisher) and purified using either mAb Select PrismA (GE Healthcare) or Capture Select (Thermo Fisher) columns, followed by SEC purification with a Superdex S200 Increase column (GE Healthcare) equilibrated with TBS (pH 8.0). For in vivo testing of IgG protective efficacy in mice, endotoxins were removed with Pierce High-Capacity Endotoxin Removal Spin Columns (Thermo Fisher), following the manufacturer's instructions. rsCSP, a recombinant, shortened construct of PfCSP containing the full N-terminal and C-terminal regions, but only 19 NANP repeats, was expressed in *E. coli* in the pET26b(+) vector, and purified as previously described[43]. Briefly, E. coli SHUFFLE competent cells were transformed with the rsCSP-pET28a plasmid, and a single colony was picked for a 50 mL overnight starter culture grown in LB broth supplemented with 50 ug/mL kanamycin. Two 1 L cultures were inoculated the next day with 25 mL each of the overnight culture, and were grown at 37 °C in LB supplemented with 50 ug/mL kanamycin. When the optical density at 600 nm reached a value of 1, the cultures were induced with 1 mM isopropyl β-D-1-thiogalactopyranoside (Sigma; cat #16758) for 6 h. The cells were then harvested and lysed by microfluidization in PBS (pH 7.4). The lysate was incubated overnight with Ni cOmplete resin (Sigma; cat # 5893682001) and was eluted in PBS (pH 7.4) containing 200 mM imidazole.

### Mutagenesis

Inferred germline sequences were identified with IgBlast and the IMGT database. Mutations in 311R, 239R, and 356R were introduced into the light chain and Fab or IgG heavy chain by mutagenic PCR, either with the QuikChange Multi Site-Directed Mutagenesis Kit (Agilent) or the Q5 Site-Directed Mutagenesis Kit (New England BioLabs). For each point mutation, the germline codon was used. Germline reversion was confirmed by Sanger sequencing.

## Sample preparation for NS and cryo-EM

Complexes of Fabs and rsCSP, or NPNA$_8$, were prepared by incubation of saturating amounts of Fab with CSP overnight at 4 °C, and purified by SEC with a Superose 6 Increase column equilibrated with TBS. For negative stain EM, complexes were diluted to ~0.05 mg/mL in TBS. The sample was applied to copper grids containing a thin film of continuous carbon, made in-house, and negatively stained with 2% uranyl formate. For cryo-EM, complexes were concentrated to 2–5 mg/mL and applied to either Quantifoil holey carbon or UltrAuFoil holey gold grids, and plunge-frozen with a Vitrobot MarkIV (Thermo Fisher).

## Negative stain electron microscopy

Room temperature imaging was performed either on a 120 keV Tecnai Spirit (Thermo Fisher) or a 200 keV Talos 200 C (Thermo Fisher) electron microscope. Datasets on the Tecnai Spirit were collected at a nominal magnification of 52,000X (2.05 Å/pix) with a Tietz TVIPS CMOS 4k x 4k camera, with a defocus of −1.5μM and a total dose of 25 e⁻/Å². Datasets on the Talos were collected at a nominal magnification of 73,000X (1.98 Å/pix) with a 4k × 4k CETA camera (Thermo Fisher), with a defocus of −1.5μM and a total dose of 25 e⁻/Å². Leginon[44] was used for automated data collection, and micrographs were stored in the Appion database[45]. Single particle analysis was performed in RELION[46], including CTF estimation, using CtfFind4[47], particle picking, and reference-free 2D classification. For 3D classification, our previous negative stain reconstruction of 311-rsCSP, low-pass filtered to 60 Å, was used as a reference. High-quality 3D classes were used as references for 3D refinement in RELION, and C1 symmetry was used in all cases.

## Cryo-EM data collection

For 227-NPNA$_8$, and 239, 334, 356, and 364 in complex with rsCSP, cryo-EM data were collected on a 200 keV Talos Arctica (Thermo Fisher) paired with a Gatan K2 Summit direct electron detector. Micrograph movies were collected at a nominal magnification of 36,000X, resulting in a pixel size of 1.15 Å, with a defocus range of −1.0 to −2.2 μm. The dose rate was ~7e⁻/pix/s for each sample, with a total of 50 frames per micrograph movie resulting in a total dose of ~50e⁻/Å². Cryo-EM data for 311 and 337 rsCSP were collected on a 300 keV Titan Krios (Thermo Fisher) with a Gatan K2 Summit direct electron detector. Cryo-EM data collection parameters for 311-rsCSP were described previously[23], and these same data were processed in this study. For 337, imaging was performed at a nominal magnification of 29,000X (1.03 Å/pix), with a defocus range of −0.9 to −2.1 μm. The dose rate was 5.3e⁻/pix/sec, and a total of 50 frames were collected resulting in a total dose of ~50e⁻/Å². In all cases, Leginon was used for automated data collection.

## Single particle Cryo-EM data processing

For 311-rsCSP, our previous cryo-EM dataset was reprocessed in the current study. Raw frames were imported into RELION3.0[48] and were aligned with the RELION implementation of MotionCor2[49]. CTF estimation was performed with CtfFind4. The Laplacian-of-Gaussian picker was used for initial autopicking on a subset of micrographs, and initial 2D templates were generated with multiple rounds of 2D classification. High-quality templates were selected as input for the automated template picker in RELION for use on the whole dataset. Multiple rounds of 2D classification were used to eliminate low-quality particles, after which a total of 605,000 particles were re-extracted for 3D classification. Our previous cryo-EM reconstruction of 311-rsCSP was used as the initial reference, low-pass filtered to 60 Å. A global angular search was used in the initial round of 3D classification, followed by multiple rounds of 3D classification without alignment. This process resulted in a final stack of ~400,000 particles that were re-extracted to generate a consensus refinement at 3.38 Å, which is the same resolution of our previous 311-rsCSP cryo-EM map generated

from these same data (EMD-9065). Further processing in RELION3.0 was used to improve the resolution of this complex. Per particle defocus values were refined in RELION, followed by another round of 3D refinement and then Bayesian polishing, which refines per-particle beam-induced motion and implements an optimized dose-weighting scheme to more accurately account for the cumulative effects of radiation damage. The resulting "shiny" particles were subjected to another round of defocus refinement and beam-tilt estimation. A final round of 3D refinement with a soft mask encompassing only the variable region of the Fabs led to the final reconstruction at 3.01 Å.

A similar protocol was followed for 356-rsCSP, using the 311-rsCSP map (low-pass filtered to 60 Å) as the initial model, leading to a 3.3 Å reconstruction in RELION3.0. The particle stack resulting from Bayesian polishing was then imported into cryoSPARCv3.3[50], and two rounds of non-uniform refinement followed by global CTF (beam-tilt) refinement was performed[51], which led to the final reconstruction at 3.2 Å.

The remaining datasets were all processed according to a similar protocol in cryoSPARC. Frames were motion-corrected with Motion-Cor2, and the aligned and dose-weighted micrographs were imported into cryoSPARCv3.3. CTF estimation was performed with CtfFind4. Autopicking was performed initially with the blob picker in cryoS-PARC, and multiple rounds of 2D classification were used to select high-quality 2D templates for subsequent template picking. Multiple rounds of 2D classification were used followed by a single round of Ab-initio reconstruction with two classes. The high-quality class was selected for further processing and was also used as the initial model. Multiple rounds of homogenous refinement, global and local CTF refinement, followed by non-uniform refinement were performed which led to the final reconstructions for each data set.

C1 symmetry was imposed for all refinements of each of the seven datasets, except for the final round of non-uniform refinement of 227, in which C2 symmetry was used. The C1 and C2 maps of 227 were nearly identical and imposing C2 improved the resolution only slightly (0.1 Å).

## Model building (cryo-EM)

For Fabs 311, 239, 356, and 364, our previously-solved X-ray structures of the corresponding Fabs in complex with NPNA$_2$ or NPNA$_3$ were used as the starting model (PDB codes 6AXK, 6W00, 6W05, and 6WFW, respectively)[18,22]. For 227, 334, and 337, an initial homology model was generated with RosettaCM[52]. For the heavy chain of each of these three Fabs, the heavy chain coordinates of the 311 X-ray structure (6AXK) were used as the template. To generate the light chain initial model, the light chain coordinates from the X-ray structure of the Fab with the corresponding light chain germline gene was used as the template: 311 for 227 (*IGLV1−40*), 239 (6W00) for 334 (*IGKV1−5*), and 356 (6W05) for 337 (*IGKV3−15*). The HC and LC templates were docked into the cryo-EM map, along with the NPNA$_2$ peptide from 6AXK, then rebuilt and refined into the map with RosettaCM and manual adjustments with Coot[53]. Individual refined Fabs were docked into the full cryo-EM map, and the CSP peptides merged into one polypeptide chain. Further manual adjustments, if necessary, were made in Coot, and the full model was refined into the density with RosettaRelax[54] and PHENIX[55].

## Structural analysis

General structural analysis, RMSD calculations, and buried surface area calculations were performed with UCSF Chimera[56]. Homotypic contacts included in Tables S3−S9 were derived from the Epitope Analyzer software, part of the ViperDB webserver[57]. Structure figures were generated with UCSF Chimera and UCSF ChimeraX[58].

## 311R X-ray structure determination

311R Fab was mixed with a 5-fold molar excess of NPNA$_3$ peptide to a final concentration of 10 mg/ml. Crystal screening was carried out

using our robotic CrystalMation high-throughput system (Rigaku, Carlsbad, CA) at The Scripps Research Institute, by vapor diffusion with 0.1 μL each of protein mixture and precipitant, with 35 μL reservoir solution. 311R-NPNA$_3$ crystals were grown in 0.04 M KH2PO4, 20% Glycerol, and 16% PEG3000 at 20°C and were cryoprotected in 30% glycerol. X-ray diffraction data were collected at the Stanford Synchrotron Radiation Lightsource (SSRL) beamline 12–1, and processed and scaled using the HKL-2000 package[59] with data reduction by POINTLESS and AIMLESS[60]. The structure was determined by molecular replacement using Phaser[61], with the 311-NPNA$_3$ X-ray structure (PDB 6AXK) as a search model. Structure refinement was performed using Refmac5[62] and iterations of refinement using Coot.

### Biolayer interferometry (BLI)
BLI was performed with the Octet Red96 (ForteBio) system. A basic kinetics experiment was used to measure binding of Fabs to NPNA$_4$, NPNA$_8$, and rsCSP. Kinetics buffer (PBS + 0.01% BSA, 0.002% Tween-20, pH7.4) was used for all dilutions, baseline measurements, and reference subtractions. Biotinylated NPNA peptides were diluted to 5 μg/mL in kinetics buffer (KB) and immobilized onto streptavidin BLI biosensors (Sartorius); His-tagged rsCSP was diluted to ~1μg/mL in KB and loaded onto Ni-NTA biosensors. Binding kinetics for each antibody were measured across a dilution series comprising the following concentrations of Fab (in nM): 6.25, 12.5, 25, 50, 100, 200. The steps of the kinetics experiment were as follows: baseline, 60 s (KB only), antigen loading, 600 s (KB + antigen), baseline 2, 60 s (KB only), association, 600 s (KB + antibody), dissociation, 1200 s (KB only). BLI data were processed with the ForteBio Data Analysis 9.0 software. In each case, global (full) fitting was performed. All curves, except for 239 R, were fit with a 2:1 binding model, as there were at least two binding sites per peptide, and these sites are likely non-independent (2 sites for NPNA$_4$, 4 sites for NPNA$_8$, and 11 sites for rsCSP). 239R exhibited 1:1 binding kinetics, and was thus best fit with a 1:1 kinetic model. For 2:1 kinetics, two $K_D$ values are reported; these are both reported in Table S10. For comparison across mAbs, and for correlation with protective efficacy, an overall affinity to each peptide was calculated as an average of these two values, which were in turn averaged across at least 4 concentrations of Fab with an $R^2$ of ≥0.98. The BLI experiments were performed in duplicate, and the final reported affinity constants were an average of the two experiments. Due to the heterogeneous nature of Fab binding to extended NPNA repeats, we refer to this value as "apparent affinity." Specifically, the two KD values reported by the Octet software likely reflect an ensemble of Fab affinities for multiple, non-independent binding sites within the same antigen molecule, as sequential Fab binding events impact the structure of the antigen (structural ordering) and expand the epitope through homotypic Fab–Fab binding.

### Liver burden assay
The protective efficacy of IgGs in this study was assessed by the reduction in liver burden assay, as previously described[22]. Mouse studies were carried out using 6–8 week-old C57BL/6 females, maintained at the animal facilities at Johns Hopkins University Bloomberg School of Public Health. Mouse rooms are kept at 40–60% relative humidity at a temperature of 68–79 degrees F, with at least 10 room air changes per hour. Mice have a cycle of 14 h light and 10 h darkness. Three separate protection experiments were conducted: one to compare the efficacy of 239R, 311R, and 356 R to WT 239, 311, and 356, and two to compare efficacy of all WT mAbs in the panel to 317. Each experiment was performed under near identical conditions. Briefly, female C57BL/6 mice, 6–8 weeks old, were injected IV with 75 μg/mouse ($N=7$) of purified IgG and sixteen hours later challenged IV with 2000 chimeric *P. berghei* sporozoites expressing *P. falciparum* CSP and, upon liver invasion, luciferase. Forty-two hours after challenge, mice were injected with 100 μl of D-Luciferin (30 mg/mL), anesthetized with isoflurane

and imaged with the IVIS spectrum to measure the bioluminescence expressed by the chimeric parasites. Characterization of the liver burden assay has demonstrated that there are not substantial differences in protective efficacy due to sex. However, to maintain consistency, only female mice were used for these experiments.

### Assessment of in vivo kinetics of IgGs
Mouse studies were carried out using 6–8-week-old C57BL/6 females, maintained at the animal facilities at The Scripps Research Institute. Mouse rooms are kept at 40–60% relative humidity at a temperature of 68–79 degrees F, with at least 10 room air changes per hour. Mice were injected IV with 75 μg of mAb per mouse. 16 h after injection, mice were bled and plasma was isolated. In parallel, a 384 well high binding plate (Corning 3700) was coated with anti-human IgG Fab antibody (Jackson ImmunoResearch 109-006-097) at a dilution of 1:500 and incubated overnight at 4 °C. The plate was blocked with 3% BSA in PBS for 1 h at RT. Plasma was added in a dilution series to the 384 well plate, and incubated for 1 h at RT. Detection was measured with alkaline phosphatase-conjugated goat anti-human IgG Fcγ (Jackson ImmunoResearch 109-005-008) at 1:2000 dilution in 1% BSA in PBS for 1 h. The plate was then washed and developed using a phosphatase substrate (Sigma-Aldrich, S0942-200TAB). Absorption was measured at 405 nm. Validation of antibodies relied on target specificity stated on the manufacturer's website; independent validation was not performed. Only female mice were used in this experiment in order to maintain consistency with the liver burden assay.

### Statistical analysis
For all liver burden experiments ($N=7$ mice), statistical significance relative to either naïve control or between experimental conditions using the measure bioluminescence flux was assessed with a Mann–Whitney $U$ test, which does not assume the data can be modeled according to a probability distribution. The data were reported as the geometric mean of the total flux in the liver ± the SD (Figs. 5, 6). This value was converted to percent inhibition relative to the naïve control, which is considered as 100% infected. Kinetic parameters from BLI experiments were derived from a non-linear regression of the reference-subtracted binding response according to a 2:1 binding model, as the immobilized antigen (rsCSP, NPNA$_8$, or NPNA$_4$) contained at least two binding sites. Values were averaged across at least 4 concentrations of Fab, and only those with $R^2 > 0.98$ were considered (Table S10). For antibody pharmacokinetics studies in mice ($N=5$), non-linear regression was used to analyze the ELISA data using Prism 9 software, and circulating human IgG concentrations were interpolated based on a standard curve; data were then reported as the geometric mean ± the SD, in mg/mL (Fig. S7).

### Reporting summary
Further information on research design is available in the Nature Portfolio Reporting Summary linked to this article.

## Data availability
Cryo-EM structures and density maps generated in this study were deposited to the Protein Data Bank (PDB) and Electron Microscopy Data Bank (EMDB), respectively, with the following accession codes: **227-NPNA$_8$:** 8DYT, EMD-27781; **239-rsCSP:** 8DYW, EMD-27784; **311-rsCSP:** 8DYX, EMD-27785; **334-rsCSP:** 8DYY, EMD-27786; **337-rsCSP:** 8DZ3, EMD-27787; **356-rsCSP:** 8DZ4, EMD-27788; **364-rsCSP:** 8DZ5, EMD-27789. The X-ray coordinates for 311 R Fab-NPNA$_3$ have been deposited to the PDB under the accession code 8EKF. The crystal structures used in this study for comparison to cryo-EM structures are available in the PDB under the following accession codes. 239-NPNA2: 6W00; 356-NPNA2: 6W05; 311-NPNA2: 6AXK; 364-NPNA2: 6WFW. Source data are provided with this paper.

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

## Acknowledgements

The authors thank B. Anderson for maintenance and administration of the cryo-EM facility at The Scripps Research Institute, and H.L. Turner and C.A. Bowman for technical support. The project was supported by National Institutes of Health grant 1F32AI150216-01A1 to G.M.M., and by funding from PATH's Malaria Vaccine Initiative and the Bill & Melinda Gates Foundation (grant INV-004923) under collaborative agreements with The Scripps Research Institute. This publication was also possible through support provided by the Office of Infectious Diseases, Malaria Branch, Bureau for Global Health, U.S. Agency for International Development (USAID), under the terms of Contract No. 7200AA20C00017. The opinions expressed herein are those of the author(s) and do not necessarily reflect the views of the U.S. Agency for International Development. Research by F.Z. and Y.F-G. is supported by the Bill and Melinda Gates Foundation (INV-001763), PATH's Malaria Vaccine Initiative and the Bloomberg Philanthropies. This research used resources of the SSRL, SLAC National Accelerator Laboratory, which is supported by the U.S. Department of Energy, Office of Science, Office of Basic Energy Sciences under Contract No. DE-AC02-76SF00515. The SSRL Structural Molecular Biology Program is supported by the DOE Office of Biological and Environmental Research, and by the National Institutes of Health, National Institute of General Medical Sciences (including P41GM103393 and P30GM133894).

## Author contributions

G.M. performed experiments, analyzed the data, prepared figures, wrote the original manuscript draft, and conceived the study. J.T., T.P., and D.O. performed experiments and analyzed the data. Y.F.G., R.M., and N.B. performed experiments, analyzed data, and wrote the manuscript. G.G., D.J., J.C., W.H.L., and G.G.P. performed experiments. D.E., R.S.M., E.L., and C.R.K. provided access to reagents and advised the study. F.Z. supervised and provided resources for the in vivo studies and analyzed the resulting data. I.A.W. and A.B.W. supervised the project, acquired funding, wrote the manuscript, and conceived the study. All authors contributed to manuscript editing.

## Competing interests

The authors declare no competing interests.
