## [Peer Review File · Nature Communications]

REVIEWER COMMENTS

Reviewer #1 (Remarks to the Author):

Martin et al. solved the cryo-EM structures of a collection of PfCSP constructs in complex with IGHV3-33 Fabs. Notably, the authors identified a spectrum of highly ordered helical Fab-CSP structures that are specific to the IGHV3-33 germline. The authors found that the CSP recognition pattern of the IGHV3-33 Fabs are quite conserved. A key finding is the helical structures are stabilized by the homotypic interactions between adjacent Fabs. Mutational analysis of the Fabs that disrupt the homotypic interactions reduced the affinity to NPNA8/rsCSP and reduced the level of protection in a transgenic mouse model for. Thus, the authors suggest that the homotypic interactions facilitate antibody affinity to extended NPNA repeats and are the key to the efficacy within IGHV-33 mAbs. Overall, the manuscript is well-written, and the assays are well-designed. The structures are solved at high resolution and the map quality is high. The authors present extensive data to validate their hypothesis derived from the structures. Overall, the manuscript is of high quality and suitable for publication if the comments below are addressed.

One concern is if the helical structures presented are representative of the *in vivo* IgG-PfCSP interaction. It would be prudent for the authors to demonstrate that full IgG molecules (not just Fabs or variable regions) can be accommodated in the helical structures identified or if there is any steric hinderance encountered by the significantly larger IgG. An analysis of the potential IgG-PfCSP structure would help convince the reader that the findings are representative of the *in vivo* interaction.

Please indicate in the structures in Figure 1 if only the variable region of the Fab is ordered, or if the entire Fab is observed.

Reviewer #2 (Remarks to the Author):

The manuscript by Martin et al., entitled "Affinity-matured homotypic interactions induce spectrum of PfCSP-antibody structures that influence protection from malaria infection" characterized how a panel of VH3-33 antibodies elicited by RTS,S recognize PfCSP. The authors describe 7 high resolution cryoEM structures of VH3-33 antibodies in complex with PfCSP or NPNA8, showing that the NANP repeat of PfCSP displays a spectrum of helical conformations mediated by homotypic interactions, many of which are via somatically hypermutated residues. The authors generated germline reverted mAbs of a selection of the panel and demonstrated a role for these SHMs for high avidity binding and protection from malaria infection in a mouse model.

Collectively, this is a well-designed and important study for the malaria vaccine field and should be of interest to the community. The cryoEM structures are well described and of excellent quality. My main concern surrounds the BLI experiments, and I have the following comments to further improve the manuscript.

- Were the BLI experiments performed more than once? It is typical to perform these measurements in either duplicate or triplicate.

- It's unclear to me why the authors perform a student's T-test to compare the affinities between antibodies and their germline reverted counterparts, assert that some of them are significantly different statistically, then state that 'binding to minimal NPNA repeats is largely unperturbed by the germline mutations' (Line 278). I would suggest that the authors remove the statistical comparison and keep their conclusion, as I agree, a two-fold difference in affinities is minimal.

- Can the authors please clarify why they refer to binding strength as apparent affinity, when Fabs and not mAbs were used in the binding studies.

- It would be beneficial to the reader to see a representative sensorgram and fit for each Fab binding to each antigen in a supplementary figure, to be able to assess the quality of the fit.

- Finally, when fitting data to a heterogenous model, typically two KDs are calculated (when using the Octet Red Data Analysis software and these should both be reported. Particularly for antibody binding to NPNA4, these two KDs would likely represent the affinity for the Fab to the antigen (lower KD) and the affinity of a second Fab to the antigen-Fab complex (higher KD). I would hypothesize that this second higher affinity interaction will likely be the one that moderately correlates with protection.

Minor Comments

The quality of the cryoEM structures described here are excellent. I just wonder what their relevance is in a vaccination setting where, in my opinion, a polyclonal response will likely prevent the ultrahelical conformations from forming on the surface of a sporozoite. Perhaps beyond the scope of this MS, but it would be beneficial to see whether the helical conformations occur in the presence of a mixture of VH3-33 mAbs. At least via negative stain. This is in contrast to the use of these VH3-33 anti-PfCSP mAbs as prophylactics, where these structures are likely more relevant.

Line 58. Certainly, the structure of the N-terminus in the context of PfCSP has not been resolved and is likely flexible, but the N-terminus does have some structure and is not completely disordered i.e. 5D5 binds to an alpha helical region of the N terminus (Thai et al., JEM 2020). Perhaps reword to flexible?

Line 89. 'potent' is italicized.

Line 104. Can the authors comment on the frequency of VH3-33 antibodies in this cohort of protected participants vaccinated with RTS,S? Are these the only seven VH3-33 antibodies sequenced or were they selected from a larger subset and if so, why?

Line 130 'seven' is italicized

Line 170. Although not in the context of rsCSP, Murugan et al., 2020 Nat Med showed a similar S-shaped conformation with six VH3-33 antibody NANP co-crystal structures and it would be appropriate to cite them here.

Line 207. In the description of the structures from line 207 onwards, FabA and FabB are swapped in Figure 3 making it hard to follow the specific interactions in the text.

Line 207. It would be beneficial to the reader to see a supplementary figure comparing the 1210-NANP5, 399-NPNA6 crystal structures with those reported here, to visualize the different conformations.

Fig 4i. 311 vs rsCSP – x-axis is missing values

Line 428. "extend" -> "extended"

Line 627. "Superose" -> "Superdex"?

Line 760 and 792. What software was used to fit the BLI data? Were they fit to a global or local fit and then averaged? It's not clear based on this description.

Reviewer #3 (Remarks to the Author):

The authors report elegant, sophisticated, well-documented studies of structural and functional analysis of multiple potent antibodies against the NANP (NPNA) repeats of the PfCSP. I think the work is

excellent. However, I question the magnitude of the additional impact in light of their 2021 publication in Nature Communications. In the previous study by many of the same authors, and using essentially the same panel of mAbs, they characterized (Nat Commun 2021 Feb 16;12(1):1063.doi: 10.1038/s41467-021-21221-4) co-crystallized structures of Fabs 239, 356, and 311 with repeat peptides (NPNA2,3,4, or 6) and found shared recognition of epitopes with secondary structural motifs in all high-affinity protective mAbs. The current studies extend the previously observed features of antibody recognition of NANP repeats, now using electron microscopy, in the same mAbs derived from the IGHV3-33 gene as they relate to functionality in the liver burden assay. My concern is also, that the authors claim (last lines of the Summary) that the findings, “advance our understanding of the mechanism(s) of antibody-mediated protection and inform next generation CSP vaccine design.” This statement has been made in perhaps 100s of papers. However, as far as I know, none have resulted in development of more potent PfCSP repeat region immunogens. Furthermore, as far as I can find, nowhere in this paper do the authors provide any indication of how they would design an immunogen that would induce more potent polyclonal antibodies against the NPNA repeats of the PfCSP. They do, however, provide ideas and some supporting data for how to do this for anti-NPNA repeat mAbs. Unless they can explicitly state and justify ideas about how they would design better immunogens, they should remove this claim from the Summary and the Discussion.

Results

Legends for Figures 5 and 6. Please provide a sentence summarizing what you think are the key findings presented in these figures, and your conclusions regarding the findings.

For polyclonal antibodies induced by vaccination with a PfCSP immunogen or anti-PfCSP mAbs to be useful in humans, they must neutralize 100% of all sporozoites to prevent blood stage infection. This is because one sporozoite can give rise to 50,000 merozoites, each of which can infect an erythrocyte. If you knock out 99 of 100 mosquito-inoculated sporozoites in a non-immune individual, a week later there will be 50,000 infected erythrocytes and 2 days later 500,000, and 2 days later symptomatic malaria. In figures 5 and 6 the maximum reduction in parasite burden was 92%. The authors are assuming that the jump from 92% to 100% is linear and dependent on the same factors that got them to 92%, but this is not proven. In papers on L9LS vs CIS43LS using this type of model, there has been the indication that L9 is significantly more potent than CIS43 based on the type of assays described herein. However, in clinical trials CIS43LS has been more protective against controlled human malaria infection than L9LS, or at least as protective. CIS43LS at 5 mg/kg SC gave 100% protection at 8 weeks against CHMI (presented at ASTMH meeting by Kirsten Lyke). L9LS has not given this level of protection. I recognize that it is easier, less expensive, and more quantitative at one level to measure liver stage burden than it is to assess total protection against blood stage infection, which is an all or none phenomenon. However, I would strongly suggest that the authors identify the minimal dose of their apparently most protective mAb that is 100% protective against blood stage infection and then assess the other mAbs at the same dose, before concluding that regarding the “most” desired outcome, which is total protection against infection, their best mAb(s) are most protective. To my knowledge there are

no data validating that reduction in liver stage burden against a chimeric Pb-PfCSP parasite in mice correlates with protection against *P. falciparum* infection in humans. Yet, this assay is used over and over again.

Discussion

The authors mention the only human mAbs which have shown protection in humans, CIS43 and L9, but fail to mention that they do not preferentially recognize the NANP (NPNA) repeats studied in this project, but rather preferentially recognize the junctional region and minor repeats between the N-terminus and NANP repeats (CIS43) and NVDP (L9). How does this fit into their conclusions or their indication that their findings will help to optimally design vaccine constructs? For the non-malaria PfCSP mAb expert this needs to be discussed. The importance of homotypic Fab-Fab interactions in protective antibodies of this class was already demonstrated in (Nat Commun 2021 Feb 16;12(1):1063doi: 10.1038/s41467-021-21221-4).

The authors mention the “extensive conformational plasticity of the NANP repeat region.” In discussing the evolution of anti-(NANP)_n mAbs, the authors never discuss, why the parasite would contain a series of repeats that are the targets of protective antibodies and at the same time are completely conserved in all *P. falciparum* isolates around the world. In other words there has been no selective pressure that has led to major variations in the repeat region sequences. Is it possible that while the sequences of (NANP)_n are conserved, the actual structures of the NANP repeat region vary (“conformational plasticity”) and that structures are “selected” among different PfCSP populations (different strains/isolates of *P. falciparum*) to be resistant to the majority of anti-NANP antibodies?

The authors state, “Our panel of cryo-EM structures will also be of immediate use in the design of NANP antigens that either promote or prevent the development of adjacent homotypic Fab-Fab interactions, or long-range bivalent IgG interactions, which may occur on extend NANP repeats.” Please explain this and how you would do it. Give examples or don’t make the claim.

REVIEWER COMMENTS

Reviewer #1 (Remarks to the Author):

Martin et al. solved the cryo-EM structures of a collection of PfCSP constructs in complex with IGHV3-33 Fabs. Notably, the authors identified a spectrum of highly ordered helical Fab-CSP structures that are specific to the IGHV3-33 germline. The authors found that the CSP recognition pattern of the IGHV3-33 Fabs are quite conserved. A key finding is the helical structures are stabilized by the homotypic interactions between adjacent Fabs. Mutational analysis of the Fabs that disrupt the homotypic interactions reduced the affinity to NPNA8/rsCSP and reduced the level of protection in a transgenic mouse model for. Thus, the authors suggest that the homotypic interactions facilitate antibody affinity to extended NPNA repeats and are the key to the efficacy within IGHV-33 mAbs. Overall, the manuscript is well-written, and the assays are well-designed. The structures are solved at high resolution and the map quality is high. The authors present extensive data to validate their hypothesis derived from the structures. Overall, the manuscript is of high quality and suitable for publication if the comments below are addressed.

One concern is if the helical structures presented are representative of the *in vivo* IgG-PfCSP interaction. It would be prudent for the authors to demonstrate that full IgG molecules (not just Fabs or variable regions) can be accommodated in the helical structures identified or if there is any steric hinderance encountered by the significantly larger IgG. An analysis of the potential IgG-PfCSP structure would help convince the reader that the findings are representative of the *in vivo* interaction.

We agree with the reviewer that IgG complex structures are the most relevant for the *in vivo* interaction. However, due to the highly repetitive nature of CSP, and the bivalent nature of an IgG, the reaction of these two components leads to rapid and complete aggregation and precipitation. We have repeatedly observed this despite multiple attempts to limit or prevent the aggregation.

In our previous publication, we were, however, able to obtain a small soluble fraction for IgG-311 in complex with rsCSP, in sufficient quantity for a negative stain reconstruction (Oyen and Torres et al., *Science Advances* 2018). This IgG-rsCSP structure closely resembled the Fab-rsCSP structure, and suggested that similar adjacent and long-range homotypic interactions can occur in the context of an IgG. *Further, the contacts identified in each of the Fab structures are highly specific, and correlate with key sites of somatic hypermutation, suggesting that these mutations evolved in the context of B-cell receptor binding to NPNA antigens in vivo. Finally, given the potential for autoreactivity these mutations are strongly and specifically driven by the CSP antigen in vivo.*

We note that the germline-reverted homotypic knockout mutants we produced (311R, 239R, 356R), were designed from Fab-rsCSP structures, yet when these antibodies are expressed as IgG, we observe a significant effect on protection *in vivo*. This suggests that these same interactions also occur in the context of IgG on the surface of sporozoites.

Collectively, these observations lead us to conclude that the Fab-rsCSP structures we have solved are biologically relevant in terms of monoclonal antibody interaction with CSP *in vivo*, either on a sporozoite or subunit vaccine.

Please indicate in the structures in Figure 1 if only the variable region of the Fab is ordered, or if the entire Fab is observed.

This is now specified more clearly in the legend for Figure 1 (Fig 1B), as follows:

“In all structures, the Fab constant domain is disordered, thus only the variable region of Fabs are modeled.”

Reviewer #2 (Remarks to the Author):

The manuscript by Martin et al., entitled “Affinity-matured homotypic interactions induce spectrum of PfCSP-antibody structures that influence protection from malaria infection” characterized how a panel of VH3-33 antibodies elicited by RTS,S recognize PfCSP. The authors describe 7 high resolution cryoEM structures of VH3-33 antibodies in complex with PfCSP or NPNA8, showing that the NANP repeat of PfCSP displays a spectrum of helical conformations mediated by homotypic interactions, many of which are via somatically hypermutated residues. The authors generated germline reverted mAbs of a selection of the panel and demonstrated a role for these SHMs for high avidity binding and protection from malaria infection in a mouse model.

Collectively, this is a well-designed and important study for the malaria vaccine field and should be of interest to the community. The cryoEM structures are well described and of excellent quality. My main concern surrounds the BLI experiments, and I have the following comments to further improve the manuscript.

- Were the BLI experiments performed more than once? It is typical to perform these measurements in either duplicate or triplicate.

The octet experiments were performed in duplicate, and the values reported are an average across the two experiments. We now state this clearly in the BLI section of the Methods (lines 861-862).

- It's unclear to me why the authors perform a student's T-test to compare the affinities between antibodies and their germline reverted counterparts, assert that some of them are significantly different statistically, then state that 'binding to minimal NPNA repeats is largely unperturbed by the germline mutations' (Line 278). I would suggest that the authors remove the statistical comparison and keep their conclusion, as I agree, a two-fold difference in affinities is minimal.

We thank the reviewer for pointing this out. We have removed statements of statistical significance for differences in affinities measured by BLI.

- Can the authors please clarify why they refer to binding strength as apparent affinity, when Fabs and not mAbs were used in the binding studies.

We thank the reviewer for drawing our attention to this important distinction. In our antigens used in the BLI experiments, npna4, npna8, and rsCSP, there are 2, 4, and 11 binding sites for this family of Fabs. As our structures show, the binding of additional Fabs after the first Fab will be engaging both the antigen and the neighboring Fab or Fabs. Moreover, the structure of CSP is affected by each sequential binding event, as an unbound and highly-flexible/disordered antigen becomes fully ordered when fully occupied. Thus, we believe that each binding event will have a distinct binding affinity. This is also demonstrated in the BLI data, as the observed affinity increases with increasing NPNA content. Given the limitations of the method, only two KDs are reported, which likely reflect the average of, in the case of rsCSP, up to 11 KDs. Thus the Kd reported is the observed, or apparent affinity, and not a true affinity of a single Fab for a single independent epitope.

We have now added a sentence in the main text, as follows (lines 305-307):

“The term ‘‘apparent affinity’’ is used to reflect the fact peptides with four or more NPNA repeats contain multiple, non-independent Fab binding sites likely with multiple, non-independent affinity constants.”

We have also expanded on our rationale for using this term in the BLI section of the Methods (lines 862-867):

“Due to the heterogeneous nature of Fab binding to extended NPNA repeats, we refer to this value as ‘‘apparent affinity.’’ Specifically, the two K_D values reported by the Octet software likely reflect an ensemble of Fab affinities for multiple, non-independent binding sites within the same antigen molecule, as sequential Fab binding events impact the structure of the antigen (structural ordering) and expand the epitope through homotypic Fab-Fab binding.”

- It would be beneficial to the reader to see a representative sensorgram and fit for each Fab binding to each antigen in a supplementary figure, to be able to assess the quality of the fit.

We have now included supplementary figures (Fig. S9) with representative binding traces and fits for all antibodies in the study.

- Finally, when fitting data to a heterogeneous model, typically two K_D s are calculated (when using the Octet Red Data Analysis software and these should both be reported. Particularly for antibody binding to NPNA₄, these two K_D s would likely represent the affinity for the Fab to the antigen (lower K_D) and the affinity of a second Fab to the antigen-Fab complex (higher K_D). I would hypothesize that this second higher affinity interaction will likely be the one that moderately correlates with protection.

The reviewer is correct that there are two K_D s reported by the Octet Red software for each Fab in complex with each peptide. To aid in the evaluation of differences, these two K_D s were averaged, and then these were averaged across each concentration in the titration. This final average then serves as the final reported K_D . We took this approach because, due to idiosyncrasies of the Octet Red Data Analysis software, K_{d1} and K_{d2} were not always consistent, i.e. the low and high affinity binding steps could be reported as either K_{d1} or K_{d2} ; thus, the designation of ‘‘ K_{D1} ’’ vs ‘‘ K_{D2} ’’ seemed arbitrary. However, across a single Fab titration, the low and high affinity K_D s were consistent, regardless of whether they were reported as K_{d1} or K_{d2} .

To better reflect this, we have now updated Table S10, which summarizes the BLI Data. We have listed both K_{d1} and K_{d2} , and the overall (average) K_D , which was used for comparison and for correlation with protection. We have also updated the BLI section of the Methods to more fully explain how the data were processed and analyzed. The added text to the methods is as follows (lines 855-8862):

“All curves, except for 239R, were fit with a 2:1 binding model, as there were at least two binding sites per peptide (2 sites for NPNA₄, 4 sites for NPNA₈, and 11 sites for rsCSP). 239R exhibited 1:1 binding kinetics, and was thus best fit with a 1:1 kinetic model. For 2:1 kinetics, two K_D values are reported; these are both reported in Table S10. For comparison across mAbs, and for correlation with protective efficacy, an overall affinity to each peptide was calculated as an average of these two values, which were in turn averaged across at least 4 concentrations of Fab with an R^2 of ≥ 0.98 . Each binding reaction was run in duplicate, and the final values reported are an average across the two experiments.”

Minor Comments

The quality of the cryoEM structures described here are excellent. I just wonder what their relevance is in a vaccination setting where, in my opinion, a polyclonal response will likely prevent the ultrahelical conformations from forming on the surface of a sporozoite. Perhaps beyond the scope of this MS, but it would be beneficial to see whether the helical conformations occur in the presence of a mixture of VH3-

33 mAbs. At least via negative stain. This is in contrast to the use of these VH3-33 anti-PfCSP mAbs as prophylactics, where these structures are likely more relevant.

We agree with the reviewer that it would be beneficial, in the context of a polyclonal immune response, to understand how multiple distinct monoclonal antibodies may simultaneously engage CSP, which may either prevent the types of homotypic interactions observed herein, or whether they promote new ones altogether. These experiments are ongoing, and are the subject of future work. Thus, as the reviewer has pointed out, we believe this goes beyond the scope of the current manuscript.

Line 58. Certainly, the structure of the N-terminus in the context of PfCSP has not been resolved and is likely flexible, but the N-terminus does have some structure and is not completely disordered i.e. 5D5 binds to an alpha helical region of the N terminus (Thai et al., JEM 2020). Perhaps reword to flexible?

Reworded "disordered" to "flexible"

Line 89. 'potent' is italicized.

Fixed

Line 104. Can the authors comment on the frequency of VH3-33 antibodies in this cohort of protected participants vaccinated with RTS,S? Are these the only seven VH3-33 antibodies sequenced or were they selected from a larger subset and if so, why?

1) Unfortunately we are unable to state specifically in this manuscript the frequency distribution of observed germlines from the RTS,S clinical trial, as this is the subject of another manuscript in process.

2) The seven vh3-33s here were selected from a larger subset based on ELISA titers to NANP6 and full length CSP. We have now specified this in the text (lines 113-114).

Line 130 'seven' is italicized

Fixed

Line 170. Although not in the context of rsCSP, Murugan et al., 2020 Nat Med showed a similar S-shaped conformation with six VH3-33 antibody NANP co-crystal structures and it would be appropriate to cite them here.

We thank the reviewer for identifying this omission, and have cited Murugan et al. 2020 here (line 179).

Line 207. In the description of the structures from line 207 onwards, FabA and FabB are swapped in Figure 3 making it hard to follow the specific interactions in the text.

We have fixed each of the Fab notations in the text to align with those shown in Figure 3 and Figure S6.

Line 207. It would be beneficial to the reader to see a supplementary figure comparing the 1210-NANP5, 399-NPNA6 crystal structures with those reported here, to visualize the different conformations.

We have now made a new supplemental figure, Figure S4, which provides an overview of the different Fab conformations observed in 356-rsCSP (representative of the VH3-33 mAbs in this study), 399-NPNA6, and 1210-NANP5.

Fig 4i. 311 vs rsCSP – x-axis is missing values

Fixed

Line 428. "extend" -> "extended"

Fixed

Line 627. “Superose” -> “Superdex”?

Fixed

Line 760 and 792. What software was used to fit the BLI data? Were they fit to a global or local fit and then averaged? It’s not clear based on this description.

We have provided more details on data analysis in the BLI section of the Methods, specifying which software was used for data processing, that global fitting was performed, that affinity values were averaged across a titration of at least four concentrations of Fab with $R^2 \geq 0.98$ (lines 849-867).

Reviewer #3 (Remarks to the Author):

The authors report elegant, sophisticated, well-documented studies of structural and functional analysis of multiple potent antibodies against the NANP (NPNA) repeats of the PfCSP. I think the work is excellent.

We thank the reviewer for such high praise!

However, I question the magnitude of the additional impact in light of their 2021 publication in Nature Communications. In the previous study by many of the same authors, and using essentially the same panel of mAbs, they characterized (Nat Commun 2021 Feb 16;12(1):1063.doi: 10.1038/s41467-021-21221-4) co-crystallized structures of Fabs 239, 356, and 311 with repeat peptides (NPNA2,3,4, or 6) and found shared recognition of epitopes with secondary structural motifs in all high-affinity protective mAbs. The current studies extend the previously observed features of antibody recognition of NANP repeats, now using electron microscopy, in the same mAbs derived from the IGHV3-33 gene as they relate to functionality in the liver burden assay. My concern is also, that the authors claim (last lines of the Summary) that the findings, “advance our understanding of the mechanism(s) of antibody-mediated protection and inform next generation CSP vaccine design.” This statement has been made in perhaps 100s of papers. However, as far as I know, none have resulted in development of more potent PfCSP repeat region immunogens. Furthermore, as far as I can find, nowhere in this paper do the authors provide any indication of how they would design an immunogen that would induce more potent polyclonal antibodies against the NPNA repeats of the PfCSP. They do, however, provide ideas and some supporting data for how to do this for anti-NPNA repeat mAbs. Unless they can explicitly state and justify ideas about how they would design better immunogens, they should remove this claim from the Summary and the Discussion.

So as to not overstate our results, we have modified the last sentence in the Summary to the following: “Together, these data emphasize the importance of anti-homotypic affinity maturation in the frequent selection of *IGHV3-33* antibodies and highlight key features underlying the potent protection of this antibody family.”

We also encourage the Reviewer to closely read our original statement in the discussion regarding vaccine design, which is as follows (lines 428-432):

“Our panel of cryo-EM structures will also be of immediate use in the design of NANP antigens that either promote or prevent the development of adjacent homotypic Fab-Fab interactions, or long-range bivalent IgG interactions, which may occur on extend NANP repeats. This will likely be invaluable in parsing the potentially countervailing forces homotypic interactions may exert on vaccine efficacy.”

We do not claim, as the Reviewer suggests above, that our structures will lead to the design of CSP vaccines which induce more potent antibodies. We state that our structures enable the structure-based vaccine design *approach*, and allow for the addressing of hypotheses raised herein, i.e. what are the effects of homotypic interactions on immunogenicity and vaccine efficacy?

We do agree with Reviewer that highlighting specific examples of how this could be done will be useful for a broad audience. Thus, we have expanded this paragraph in the Discussion (lines 475-483). Again, our emphasis is on how our data inform the rational vaccine design approach itself, not necessarily that they tell us how to make more potent immunogens. The expanded discussion is as follows: “Our panel of cryo-EM structures will enable the rational design of NANP antigens that either promote or prevent the development of adjacent homotypic Fab-Fab interactions, or long-range bivalent IgG interactions, which may occur on extended NANP repeats. For example, based on our cryo-EM structures, a full helical turn encompasses a minimum of 10 and maximum of 12 NPNA repeats, as observed in 337-rsCSP and 239-rsCSP, respectively. Thus, a candidate antigen containing less than 10 NPNA repeats will prevent development of the long-range homotypic interactions which stabilize extended CSP helices, which could impact the maturation pathway of induced antibodies. Conversely, the long-range CSP helical structure could be biochemically stabilized in a designed construct to specifically elicit highly potent 311, 356, or 239-like antibodies. Long-term, comparing a series of such antigens will be useful in parsing the potentially countervailing forces homotypic interactions may exert on vaccine efficacy.”

Results

Legends for Figures 5 and 6. Please provide a sentence summarizing what you think are the key findings presented in these figures, and your conclusions regarding the findings.

We thank the Reviewer for this suggestion. We have now included such statements in Figure legends 5 and 6.

For polyclonal antibodies induced by vaccination with a PfCSP immunogen or anti-PfCSP mAbs to be useful in humans, they must neutralize 100% of all sporozoites to prevent blood stage infection. This is because one sporozoite can give rise to 50,000 merozoites, each of which can infect an erythrocyte. If you knock out 99 of 100 mosquito-inoculated sporozoites in a non-immune individual, a week later there will be 50,000 infected erythrocytes and 2 days later 500,000, and 2 days later symptomatic malaria. In figures 5 and 6 the maximum reduction in parasite burden was 92%. The authors are assuming that the jump from 92% to 100% is linear and dependent on the same factors that got them to 92%, but this is not proven. In papers on L9LS vs CIS43LS using this type of model, there has been the indication that L9 is significantly more potent than CIS43 based on the type of assays described herein. However, in clinical trials CIS43LS has been more protective against controlled human malaria infection than L9LS, or at least as protective. CIS43LS at 5 mg/kg SC gave 100% protection at 8 weeks against CHMI (presented at ASTMH meeting by Kirsten Lyke). L9LS has not given this level of protection. I recognize that it is easier, less expensive, and more quantitative at one level to measure liver stage burden than it is to assess total protection against blood stage infection, which is an all or none phenomenon. However, I would strongly suggest that the authors identify the minimal dose of their apparently most protective mAb that is 100% protective against blood stage infection and then assess the other mAbs at the same dose, before concluding that regarding the “most” desired outcome, which is total protection against infection, their best mAb(s) are most protective. To my knowledge there are no data validating that reduction in liver stage burden against a chimeric Pb-PfCSP parasite in mice

correlates with protection against *P. falciparum* infection in humans. Yet, this assay is used over and over again.

We thank the reviewer for their comments. However, we disagree with their interpretation of our results and of the utility of the liver burden endpoint for our study.

First, to our knowledge, there is no head-to-head comparison of L9LS and cis43LS with any model; if the reviewer knows of such a paper, we would appreciate being pointed to it. This is important as the LS versions may yield different results than non-LS. In terms of cis43 and L9, the liver burden data correspond well with the mosquito bite challenge in mice (Wang et al. *Immunity* 2020; Kialu et al. *Nat Med* 2018). However, in humans, there are insufficient data to claim that cis43LS is at least as protective or better than L9LS. There are now three studies that look at protection of mAbs in humans, namely cis43LS (Gaudinski et al. *NEJM* 2021 and Lyke et al. *Lancet Inf Dis* 2023) and L9LS (Wu et al. *NEJM* 2022). For cis43LS, it was demonstrated that 4 of 4 individuals who received 5 mg/kg of cis43LS intravenously or subcutaneously had sterile protection after CHMI (parasite-free by PCR for 21 days). The results are almost identical for L9LS: 4/4 participants receiving 5 mg/kg intravenous were sterilely protected. However, 4/5 participants receiving 1 mg/kg L9LS intravenously were also sterilely protected, while 4/7 participants receiving 1 mg/kg cis43LS intravenously developed parasitemia. These data suggest the opposite conclusion from that of the Reviewer, that L9LS is at least if not more protective than cis43LS, which would correspond with both liver burden and parasitemia mouse model endpoints. Given the difficulty of clinical trials in humans, the fact that there are no human experiments directly validating or replicating results from the Pb-PfCSP mouse model, or any mouse protection assay, does not invalidate the use of the mouse model altogether. Rather, the human cis43LS and L9LS trials, in our opinion, validate the Pb-PfCSP liver burden experiment, as this assay was used to identify these mAbs as highly protective among large panels of isolated mAbs, and in humans they have also shown to be highly protective.

Second, our use of the Pb-PfCSP liver burden experiments was to simply compare the relative potency of the mAbs in our panel. We do not claim that 92% reduction of parasites in the liver means sterile protection, nor is our goal, or “most desired outcome” to demonstrate that these mAbs can provide sterile protection in humans, or are more or less protective than any other mAbs in any other paper. Specifically, we are interested in the roles of homotypic interactions in the mechanism of protection, and how affinity contributes to potency. As has been demonstrated (Raghunandan et al. *Malaria J* 2020), the liver burden and parasitemia assays are highly consistent, and the inherently quantitative readout of the liver burden assay is amenable to detecting differences in potency in mAbs. Thus, we believe that in this study the liver burden endpoint is well-suited to address our research questions.

Finally, the Reviewer has proposed we do a series of parasitemia experiments to allow us to make the claim that “our best antibodies are the most protective;” namely, that we should titrate our best mAb or mAbs to find the lowest dose for sterile protection, and then repeat the experiment at this dose for all the 11 mAbs in our panel. While compelling, these *in vivo* experiments extend well beyond the scope of our paper and would also be very costly and time-consuming to complete. As we have argued above, the liver burden assay corresponds well with mosquito bite challenge and is also well-suited for comparing relative potency across a panel of mAbs. As the Reviewer has pointed out, the mosquito bite challenge assay is “all-or-none”, as only a single parasite can lead to a full-scale blood stage infection. This produces more stochastic data, and is thus more challenging to identify statistical differences. Thus, overall, we do not believe these additional experiments will provide substantially more or novel insight to the manuscript.

Discussion

The authors mention the only human mAbs which have shown protection in humans, CIS43 and L9, but fail to mention that they do not preferentially recognize the NANP (NPNA) repeats studied in this project, but rather preferentially recognize the junctional region and minor repeats between the N-terminus and NANP repeats (CIS43) and NVDP (L9). How does this fit into their conclusions or their indication that their findings will help to optimally design vaccine constructs? For the non-malaria PfCSP mAb expert this needs to be discussed. The importance of homotypic Fab-Fab interactions in protective antibodies of this class was already demonstrated in (Nat Commun 2021 Feb 16;12(1):1063doi: 10.1038/s41467-021-21221-4).

We agree with the reviewer that it is important to point out that these are non-NPNA mAbs. We have now specified this in the discussion, and further explained the significance of these results for the development of mAbs prophylactics in general (lines 450-454).

We do not believe that the cis43LS and L9LS human trials are directly relevant to the usefulness of our structural data for vaccine design. We are discussing these mAbs in the context of rational antibody engineering, as introduced at the top of the paragraph (lines 445-450), and for which structures of antibodies, like the ones we present in the manuscript, are a necessary pre-requisite. We then elaborate on how this can be done, as with cis43 (Kratovich et al. Immunity 2021).

In terms of the previous publication on this antibody family (Pholcharee et al. Nat Comms 2021), it was demonstrated that 239 utilizes homotypic interactions by X-ray crystallography, and that this antibody and others in the family display a substantial increase in avidity to NPNA peptides with increasing NPNA content. However, it was not directly demonstrated that homotypic contacts underlie this phenomenon, and thus the “the importance of homotypic Fab-Fab interactions in protective antibodies of this class” was not directly addressed, as the Reviewer suggests. This question is one of the main points of the current manuscript.

The authors mention the “extensive conformational plasticity of the NANP repeat region.” In discussing the evolution of anti-(NANP)_n mAbs, the authors never discuss, why the parasite would contain a series of repeats that are the targets of protective antibodies and at the same time are completely conserved in all *P. falciparum* isolates around the world. In other words there has been no selective pressure that has led to major variations in the repeat region sequences. Is it possible that while the sequences of (NANP)_n are conserved, the actual structures of the NANP repeat region vary (“conformational plasticity”) and that structures are “selected” among different PfCSP populations (different strains/isolates of *P. falciparum*) to be resistant to the majority of anti-NANP antibodies?

We thank the Reviewer for bringing to light this very interesting topic. We agree that the conformational plasticity of PfCSP may have fundamental implications for both the selection and maturation of antibodies and their ability to block infection. We have added a paragraph at the end of the Discussion to address this, which is as follows:

“It is possible that the conformations of rCSP captured here by cryo-EM correspond to distinct conformations of PfCSP on RTS,S particles or sporozoites, and that the maturation pathway of this antibody family proceeded through conformational selection of these different states. Intriguingly, while the number and positioning of the NANP repeats can vary across *P. falciparum* isolates, the amino acid sequence (NANP) is highly conserved⁴¹, which suggests that the intrinsic disorder of the repeat region itself may serve as an immune evasion technique. Thus, the relative rarity of highly potent anti-PfCSP antibodies may be a function of the rarity of

PfCSP conformational states competent to bind them. Further structural characterization of PfCSP, both in complex with antibodies and in the native context of sporozoites, will be critical for understanding the complex ways in which *P. falciparum* can interact with and evade the human immune system.”

The authors state, “Our panel of cryo-EM structures will also be of immediate use in the design of NANP antigens that either promote or prevent the development of adjacent homotypic Fab-Fab interactions, or long-range bivalent IgG interactions, which may occur on extend NANP repeats.” Please explain this and how you would do it. Give examples or don’t make the claim.

We believe that we have adequately addressed this topic above, in the first comment from the Reviewer, by adding specific examples in the Discussion (lines 475-483).

REVIEWERS' COMMENTS

Reviewer #1 (Remarks to the Author):

The authors have addressed all my comments and the revised manuscript is much improved.

Reviewer #4 (Remarks to the Author):

The work is excellent and the manuscript clearly written. The authors have taken into account most remarks from the reviewers, including reviewer #3. In particular, they discuss in greater details the relevance of their findings in the context of “real” polyclonal Ig responses versus monoclonal antibody therapies.

My main criticism (shared with reviewer #3) is that conceptually this study provides limited additional insights as compared to their previous publications (in particular Pholcharee et al, Nature Communications 2021), where the role of homotypic Fab-Fab interactions in antibody binding to CSP and in protection was already addressed. Similarly, the importance of anti-homotypic affinity maturation in the frequent selection of IGHV3-33 antibodies has been reported before.

However, in the present manuscript, the authors directly demonstrate the importance of homotypic interactions through mutagenesis of the homotypic interface. The data convincingly show that impairing homotypic interactions through reversion to the germline sequence results in a reduction of the avidity of Fab binding to extended NANP repeats (fig 4), and, most importantly, in lower protective efficacy of the corresponding mAbs (fig 5). For these experiments, the authors used a highly relevant mouse infection model based on transgenic *P. berghei* parasites expressing PfCSP and luciferase. While the results presented in fig 6 show a correlation between the apparent affinity of Fabs and the inhibitory activity of mAbs, the question remains whether homotypic interactions are relevant in the context of entire IgG molecules and critical for the neutralizing potential of the anti-CSP mAbs. This is probably a difficult question to address, given the technical limitations with using mAbs, as clearly explained by the authors in their manuscript.

REVIEWERS' COMMENTS for NCOMMS-22-45137A

Reviewer #1 (Remarks to the Author):

The authors have addressed all my comments and the revised manuscript is much improved.

We thank the Reviewer for their helpful comments and suggestions for our manuscript.

Reviewer #4 (Remarks to the Author):

The work is excellent and the manuscript clearly written. The authors have taken into account most remarks from the reviewers, including reviewer #3. In particular, they discuss in greater details the relevance of their findings in the context of “real” polyclonal Ig responses versus monoclonal antibody therapies.

My main criticism (shared with reviewer #3) is that conceptually this study provides limited additional insights as compared to their previous publications (in particular Pholcharee et al, Nature Communications 2021), where the role of homotypic Fab-Fab interactions in antibody binding to CSP and in protection was already addressed. Similarly, the importance of anti-homotypic affinity maturation in the frequent selection of IGHV3-33 antibodies has been reported before.

However, in the present manuscript, the authors directly demonstrate the importance of homotypic interactions through mutagenesis of the homotypic interface. The data convincingly show that impairing homotypic interactions through reversion to the germline sequence results in a reduction of the avidity of Fab binding to extended NANP repeats (fig 4), and, most importantly, in lower protective efficacy of the corresponding mAbs (fig 5). For these experiments, the authors used a highly relevant mouse infection model based on transgenic *P. berghei* parasites expressing PfCSP and luciferase. While the results presented in fig 6 show a correlation between the apparent affinity of Fabs and the inhibitory activity of mAbs, the question remains whether homotypic interactions are relevant in the context of entire IgG molecules and critical for the neutralizing potential of the anti-CSP mAbs. This is probably a difficult question to address, given the technical limitations with using mAbs, as clearly explained by the authors in their manuscript.

We thank the Reviewer for their comments. We agree that it is critical to understand the role of homotypic interactions in the context of full IgG molecules. However, given the technical difficulty in studying this interaction *in vitro*, due to rapid and complete aggregation of the IgG-CSP complex, these experiments are beyond the scope of this manuscript, and are the subject of further, ongoing investigation.